# TOWARDS DISTRIBUTED NEURAL ARCHITECTURES

## ABSTRACT

We introduce and train distributed neural architectures (DNA) in vision and language domains. DNAs are initialized with a proto-architecture that consists of (transformer, MLP, attention, etc.) modules and routers. Any token (or patch) can traverse any series of modules in any order. DNAs are a natural generalization of the sparse methods such as Mixture-of-Experts, Mixture-of-Depths, parameter sharing, etc. Computation and communication patterns of DNA modules are learnt end-to-end during training and depend on the content and context of each token (or patch). These patterns can be shaped by further requirements added to the optimization objective such as compute/memory efficiency or load balancing. We empirically show that (i) trained DNAs are competitive with the dense baselines in both domains and (ii) compute efficiency/parameter sharing can be learnt from data. Next, we analyze the emergent connectivity and computation patterns in the trained DNAs. We find that the paths that tokens take through the models are themselves distributed according to a power-law. We show that some paths (or, equivalently, groups of modules) show emergent specialization. Finally, we demonstrate that models learn to allocate compute and active parameters in an interpretable way.

## 1 INTRODUCTION

Over the last few years, scale has been the main driver of the progress in AI. The most reliable recipe to train a better (base) model has been to increase the model size and the amount of data[1]. Furthermore, with the advance of the transformer architecture it became clear that GPU utilization is one of the key factors that determines the quality of a model because more compute translates into better performance Kaplan et al. (2020); Achiam et al. (2023); Grattafiori et al. (2024). Consequently, architectures that leverage parallel execution of matrix multiplication became dominant Vaswani et al. (2017). As AI workloads are becoming increasingly wide-spread in all domains we expect a steep rise of the inference compute expenditure. Consequently the task of developing methods that save inference compute is critical. There is a large body of work in efficiency including distillation Hinton et al. (2015), pruning LeCun et al. (1989), quantization Jacob et al. (2018), over-training Sardana et al. (2023), model-routing Chen et al. (2023), speculative decoding Leviathan et al. (2023), as well as mixtures of these techniques Muralidharan et al. (2024).

These challenges motivate developing distributed neural architectures (DNA) Fig. 1. These architectures are not feed-forward and allow information to flow between any pair of computing modules Figs. 1- 2. Connectivity of DNAs emerges from end-to-end training Figs. 3, 4, 13, 8. Finally, DNAs are trained to allocate compute dynamically based on the input data Fig. 5. Another motivation for this work comes from the recent work on layer pruning Gromov et al. (2024), Csordás et al. (2025), where it was found that deeper layers contribute unreasonably little to certain benchmarks. Our main inspiration is the work on *conditional computing* (CC) Bengio et al. (2013), of which MoE is the prime example. There, different tokens (or patches) are allowed to take different paths through the network. Each path is decided by an additional structure called the *router*. Today, the main application of CC is (partial) decoupling of parameter count and compute by *activating* only a fraction of parameters per forward pass Shazeer et al. (2017) (and Jacobs et al. (1991)). Routing can also be used to control the amount of compute (or, equivalently, the number of active parameters) *per token*. The main example of this idea is Mixture-of-Depths (MoD) Raposo et al. (2024) and Layer-skip Elhoushi et al. (2024) (which is based on self-speculative decoding).

---

[1]Assuming that the larger model is trained intelligently and the extra data is diverse and good quality.

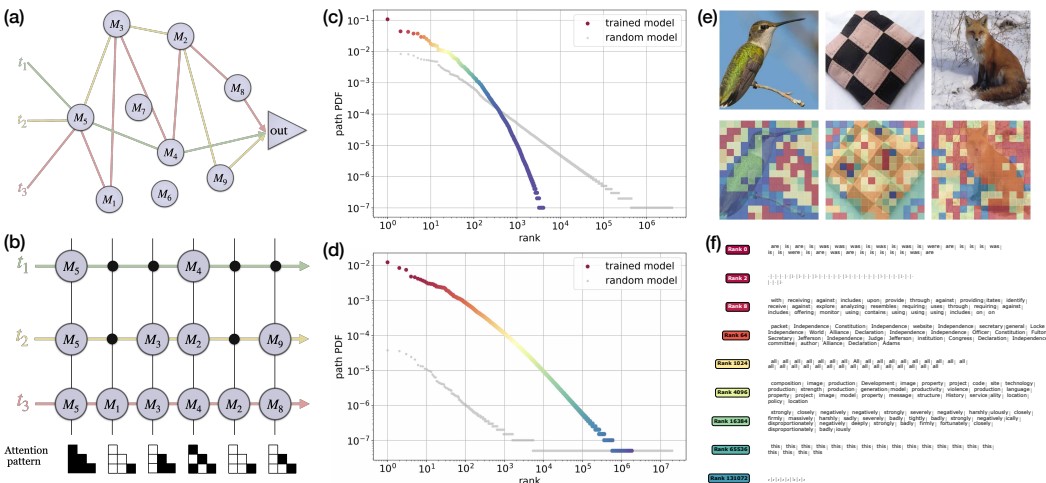

Figure 1: **(a)** DNA from the module's perspective: Each token follows its own trajectory through the network. There is no notion of depth or width. When the objective function includes compute efficiency some paths are shorter than others. **(b)** DNA from the token's perspective: The forward pass consists of a series of steps. At any given step, each token is acted on by a module or by an identity operation (black dots). When a module that contains attention operation acts on several tokens simultaneously the attention pattern is computed *only* between these tokens. Consequently, the attention part of the architecture can be understood as dynamic (*i.e.* data-dependent) sparsity. **(c)** Path distribution of top-1 trained DNA vision model: we plot the distribution of paths that tokens take through the model collected from test split of ImageNet. The color corresponds to the frequency of the paths and is used in the sub-figure **(e)**. Surprisingly, the distribution of paths through the random model also follows power-law with exponent $-1$. **(d)** Path distribution of top-1 trained DNA language model: we plot the distribution of paths that tokens take through the model. The color corresponds to the frequency of the paths and is used in the sub-figure **(f)**. The distribution is a power-law with exponent $-1.2$. Similarly, we find that paths through the random model are distributed according to a power-law with exponent $-1$. **(e)** We highlight the paths that different patches take through the DNA by their frequency. We see very clear *emergent* specialization of paths: some paths focus on the object, others on the background and others on the edges and boundaries. This visualization also illustrates the sparse nature of the attention: patches of different colors do not attend to each other in later steps of the forward pass. **(f)** Examples of tokens that follow frequent (low rank) and rare (high rank) paths. We see that paths specialize in different ways: versions of the verb "to be", embedding of "." which we view as sentence-level attention, adjectives, etc.

In this work, we leverage conditional computing to create DNA models whose connectivity emerges from the end-to-end training. DNAs are initialized with a *proto*-architecture that consists of a collection of routers and a collection of modules such as MLP, attention, transformer, etc[2]. Each token is routed via its own path through the network. Modules and routers are optimized jointly. This construction includes feed-forward, MoE, MoD, weight sharing, early exit as particular cases that can emerge via optimization. In practice, we find that a mixture-of-*all*-of-these-methods emerges from end-to-end training.

We train two types of models: (i) discriminative models in vision domain trained on ImageNet Deng et al. (2009) and (ii) generative NLP models trained on a subset of fineweb-edu. Our main findings are[3]

- In both domains DNA models are competitive with dense baselines.
- DNA models can learn to use less compute with minor effects on performance.
- The paths taken by the tokens/patches and routing decisions are often human-interpretable.

---

[2]We have not yet experimented with including other modules

[3]We emphasize that our work is *not* focused on beating SOTA models in any domain, but on showing that distributed models are *feasible* and on analyzing their emergent structure.

- Individual modules, groups of modules, as well as paths specialize.

The paper is organized as follows: in Section 2 we explain the general setup, establish some terminology, and introduce certain choices we made to ensure trainability. In Section 3 we present the vision DNA models and analyze their inner-workings, while in Section 4 we do the same for the generative language DNA models. In Section 5 we present our conclusions and discuss the new emerging research directions.

## 2 DISTRIBUTED NEURAL ARCHITECTURES

### 2.1 GENERAL SET UP AND INTUITION

In this Section we describe the general structure of DNAs, as well as different ways of reasoning about its nature. The main idea can be traced back to the classic work of Minsky (1986): we view a neural network as a collection of computational modules that develop specialization, learn interaction and composition end-to-end. The internal structure of the computational modules as well as communication pattern between them are *emergent*. Our objective is to train well-performing models and then to understand the distributed nature of computation, emerging connectivity and specialization rather than saturating any particular benchmark. In order to take full advantage of DNAs the infrastructure has to be co-designed with the emergent structure (or, equivalently, infrastructure should shape/constrain the emergent structure). We leave this direction to the future work.

**General DNAs**. In the general setting, the proto-architecture of DNAs is a collection of the following components (i) Input node (including embedding/patchify layers), (ii) Output node (including unembedding layer), (iii) $N_m$ distinct computational modules, $M_i$, (iv) $N_r$ distinct routers, $R_j$.

Each computational module operates on sequences and can be a transformer, MLP, attention, mamba, convolution or any other operation. The total number of modules $N_m$ determines the total number of parameters, but not the compute. Each router is a classifier with (at most) $N_m$ classes. Routers process tokens in parallel and send them out to the corresponding modules. The router can be trained with a general top-$k$ choice. In our experiments we take $k = 1, 2$.

The forward pass (after traditional embedding operations) works as follows: (i) token embeddings are routed to relevant modules, (ii) modules perform computation on token embeddings they received, (iii) recomputed embeddings are put back together and step (i) is repeated. In Fig. 1 we show the forward path from the perspective of modules and from the perspective of tokens. The latter makes it clear that the forward pass is fully causal (in contrast to MoD Raposo et al. (2024)), while the former emphasizes distributed nature of DNA.

### 2.2 TECHNICAL DETAILS

In this paper, we made a few purely empirical design choices that allowed to take advantage of optimizations such as flash attention as well as to reduce the search space. Each of our modules can be chosen from the classic GELU-transformer block or its attention/MLP component. In all cases, we use the Pre-LayerNorm (Pre-LN) version. Many improvements/gains are certainly left on the table.

**Routing.** Our routers are linear (token-choice) classifiers. The probabilities of selecting a given $M_i$ at step-$s$ for token $t$ is obtained by applying softmax over router logits, where the routing decision is made by sampling with hard top-$k$. This softmax results in $\rho^{(s,t)} = \text{softmax}(R_s(\boldsymbol{h}^{(s,t)}))$. Each token chooses $k$ modules to participate in, leading the $i^{\text{th}}$ module to have an input, $\boldsymbol{h}_i^{(s)}$, which is the collection of those tokens. At each step-$s$, the outputs of the $k$ chosen modules are combined as follows[4]

$$\boldsymbol{h}^{(s+1,t)} = \boldsymbol{h}^{(s,t)} + \sum_{i \in \text{top-}k_\star(\rho_\star^{(s)})} \rho_i^{(s)} \left( M_i^t(\boldsymbol{h}_i^{(s)}) - \boldsymbol{h}^{(s,t)} \right), \tag{1}$$

---

[4]Somewhat awkward for of Eq.1 is explained by the fact that $M_i^t(\boldsymbol{h}_i^{(s)})$ is assumed to have a skip connection built-in, so we subtract $\boldsymbol{h}^{(s,t)}$ and the add it later to make sure we do not overcount it and end up with blowing up signal.

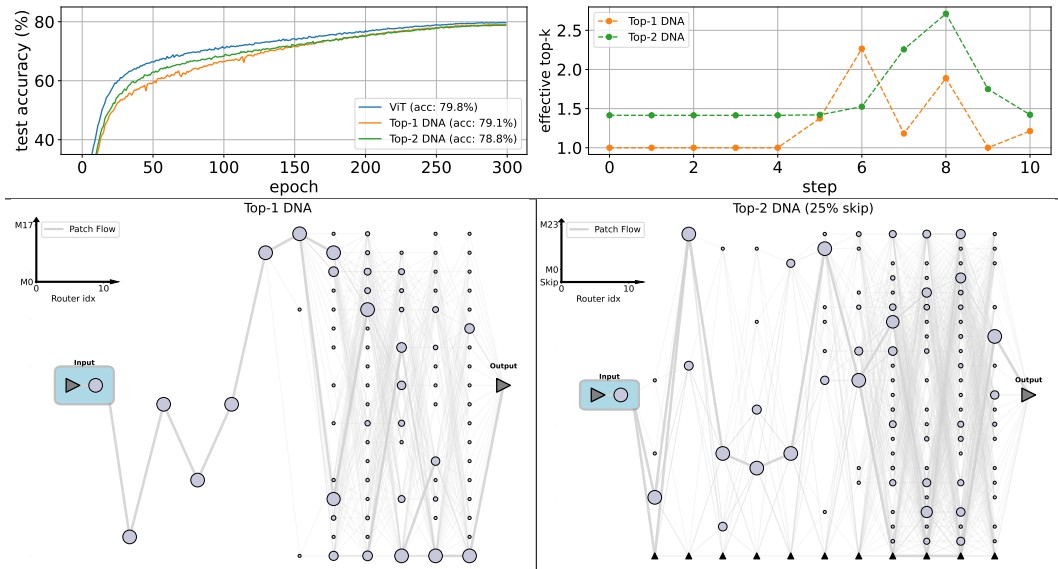

Figure 2: **Top-Left**: Test accuracy on ImageNet validation set during training for DNAs and the dense baseline. **Top-Right** Effective number of compute nodes used per step, for DNA models at the end of training. In both cases, the models exhibit higher diversity in routing decisions near the output. **Bottom**: Flow of patches plotted using the validation set of the ImageNet for the top-1 (left) and top-2 (right) models. Each column of circles contains *all* modules in the DNA. The size of the circles is proportional to frequency with which each module is activated. If a circle is absent, then it is never activated. We see that the models develop a dense foundation and later split into distributed sparse networks.

where $\boldsymbol{h}^{(s)}$ is the pre-activation at step-$s$ and $\star$ denotes the collection of all possible module indices, and the $t$ superscript on $M$ denotes the $t^{\text{th}}$ component of the output. This specific choice is made building on Roberts et al. (2022); Doshi et al. (2023) to ensure good signal and gradient propagation.

The routers are arranged as follows: at each token-processing step $s$ a router $R_s$ makes the decision for *all* tokens. The total number of token processing steps is capped by a hyperparameter $s_{\max}$ which determines the maximum compute per token. While this is not the most general way of arranging the routers, we had to limit our exploration to converge in a finite time. Finally, we found that the optimization converges much better if first few modules are not routed, meaning they process all tokens and that choice is hard-coded. We refer to these few layers as *backbone* and denote the number of layers in the backbone as $N_b$. In our experiments $N_b = 0, 1, 2$.

**Emergent compute efficiency.** In order to teach the DNA to use compute intelligently we introduce several *identity* modules that do not apply any operation: If a router sends a token to the identity module then nothing happens to the token ($\boldsymbol{h}^{(s+1)} = \boldsymbol{h}^{(s)}$). To encourage the network to skip modules (and save compute) we follow the bias trick from DeepSeek model Liu et al. (2024): We introduce one bias $b_i^{(s)}$ for each step $s$. These biases are decoupled from the Autograd as in Liu et al. (2024). While in the original work the biases were used to enforce load balancing we use them to encourage the models to use identity layers by modifying the top-$k$ selection part of the forward pass via

$$i \in \text{top-}k \left( \rho_\star^{(s)} + b_\star^{(s)} \right) . \tag{2}$$

Note the biases are only non-zero when the index $i$ corresponds to Identity module and will not affect the probability in Fig. 1.

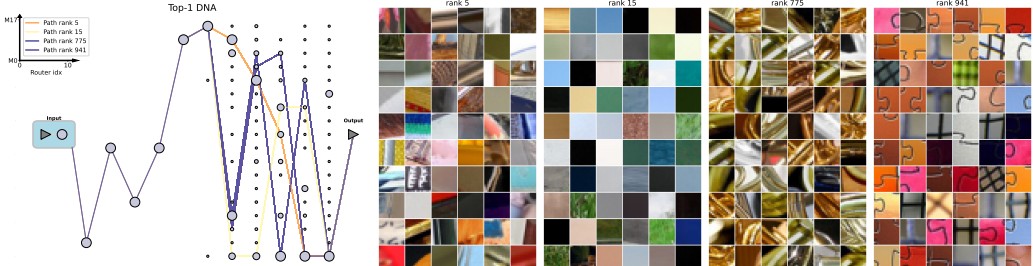

Figure 3: **Left**: Paths followed by the patches on the right. The color corresponds to the rank. **Right**: Patches that follow the highlighted paths. The patches that pass through lower-rank paths share a higher-level commonality (color and edges), while patches that go through higher-rank paths are associated with the more specific concept (brass instruments and puzzle pieces).

We initialize all biases at $0$ and only update the term corresponding to identity modules during training

$$b_i^{(s)}(t+1) = b_i^{(s)}(t) + u \cdot \text{Sign}\left(r \cdot k \cdot \bar{c}^{(s)}(t) - \sum_{i \in \text{Id}} c_i^{(s)}(t)\right), \qquad (3)$$

where $c_i^{(s)}$ represents the token counts that pass through a given module, and $\bar{c}^{(s)}$ denotes the average token counts across all modules, $r$ is introduced to control the ratio of tokens to skip and $u$ is for controlling the update speed of the bias term. We emphasize that because of the identity modules each token is processed using different amount of compute. Furthermore, the amount of compute per sequence varies due to sparse attention patterns.

**Optimization objective.** We optimize a traditional categorical cross-entropy loss function with AdamW optimizer with learning rate warm-up and cos-decay for vision and warm-up stable decay for language. The values of hyperparameters, initialization scheme, etc can be found in the Fig. A. We do *not* use load-balancing because our objective is to let models develop the structures they need in order to solve the task and then to understand these structures. Optimization of DNAs for real-world inference is delegated to the future work.

| Model | $N_b$ | $N_m$ | $N_r$ | $d_{\text{embed}}$ | $d_{\text{MLP}}$ | $N_{\text{head}}$ | Active Params | Params |
|---|---|---|---|---|---|---|---|---|
| ViT-small | - | 12 | - | 384 | 1536 | 6 | 22M | 22M |
| top-1 DNA | 1 | 18 | 11 | 384 | 1536 | 6 | 22M (17M) | 34M |
| top-2 DNA (25% skip) | 1 | 24 | 11 | 256 | 1024 | 4 | 18M (15M) | 18M |

Table 1: Hyperparameters of DNA models and ViT-small. Note that for DNA models, the number of active parameters shown in parentheses refers to non-shared active parameters (as detailed in Sec. 3.3).

## 3 DNA FOR COMPUTER VISION

### 3.1 EXPERIMENTAL DETAILS

We trained several DNA models at a ViT-Small scale (Dosovitskiy et al., 2021), with the complete set of hyperparameters reported in Fig. 1. In all cases, we used the same data-augmentation pipeline—RandomCrop, RandomHorizontalFlip, and ColorJitter (Krizhevsky et al., 2012); AutoAugment (Cubuk et al., 2019); Random Erasing (Zhong et al., 2020); MixUp (Zhang et al., 2018); and CutMix (Yun et al., 2019). All models were trained for 300 epochs with a batch size of 2048. We performed a grid search over learning rates $\eta \in \{0.001, 0.0015, 0.002\}$ and weight-decay values $\lambda \in \{0.02, 0.05, 0.1, 0.2\}$. For further details, we refer the readers to Appendix A. The training curves for the best run of each model and the routing patterns of the top-1 and top-2 DNA models are shown in Fig. 2.

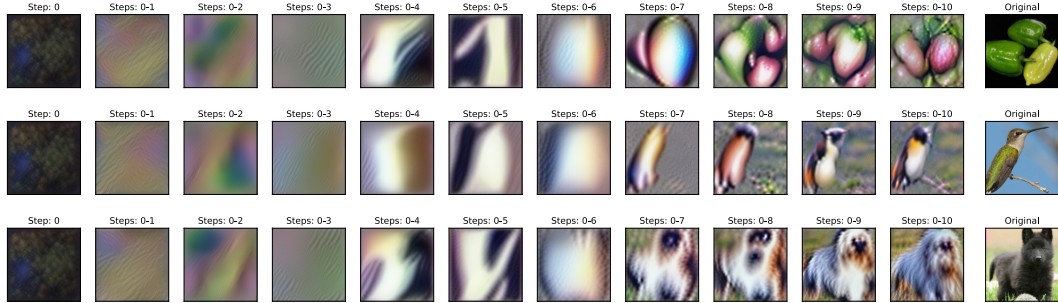

Figure 4: Three examples of reconstructed images: bell pepper, hummingbird, and Welsh springer spaniel (top to bottom). Images are reconstructed by maximizing the total weight on *all* routing decisions made from step 0 to step $i$ of the forward pass. We can see that the early layers (1-3) are primarily concerned with texture and edges, intermediate layers (4-6) with lighting distributions, and the remainder (7-10) host larger-scale features. The final reconstructed images (top to bottom) are classified as "spotlight" ($p = 0.48$), "hummingbird" ($p = 0.55$), and "papillon" ($p = 0.44$). In the last two cases, all top 5 model guesses are birds and dog breeds correspondingly, reflecting hierarchical nature of ImageNet. Model guesses for first row are: spotlight, matchstick, candle, car mirror, snowmobile.

## 3.2 INTERPRETABILITY

**Patches and Paths.** The paths taken by patches are highly interpretable. To demonstrate this, we used the top-1 DNA model shown in Fig. 2 and collected the paths of all patches from images in the ImageNet validation set. In Fig. 3, we highlight four representative paths (colored by their corresponding CDF values), along with 60 randomly selected patches that follow each path.

We observed that frequently taken paths (i.e., those with low rank) tend to aggregate patches from diverse images that nevertheless share high-level features—for example, edges in the case of the rank-5 path, and flat color regions for the rank-15 path. In contrast, infrequently taken paths (i.e., those with high rank) tend to group patches from visually similar images with low-level similarities—for instance, brass instruments for the rank-775 path and puzzle pieces for the rank-941 path. To establish a baseline, we performed a similar analysis on a randomly initialized DNA model. Somewhat surprisingly, we found that it can also cluster images. However, it uses a very different similarity measure, leading to nearly identical patches being clustered based on superficial features[5]. See Appendix G.2 for further discussion.

We have also found that in the cases when an image has a clearly defined object, such as in Fig. 1 and Fig. 13, there is a group of patches that follows the boundary between object and the background. In fact, the most compute-heavy images are the ones with very intricate collection of boundaries Fig. 5. These observations appear related to the work Riquelme et al. (2021) where it was found that the patches most critical for classification are of the same nature.

**Routing decision visualization.** To further understand the DNA routing decision structure we generate the images whose patches take exact same paths as a given image from ImageNet. Concretely, each image contains 192 patches and each patch follows a patch described by 12 integers (routing decisions, top-1 case). Each step of forward path is described by 192 integers – routing choices for each patch. For $k$ steps we specify $k \cdot 192$ integers. We use deep-dream method to generate an image that that travels the same 192 paths as the original image with highest probability. Fig. 4 depicts how the image develops when the patches match more steps of the paths of original images. Early decisions are not informative because the network is dense for the first few steps. Then human-interpretable features rapidly develop as deeper routing decisions are maximized. The final image looks recognizable, but not the same as the original. Furthermore, the network is guess the group of classes correctly: birds and dogs, but has difficulty deciding the species and breed. The onset of interpretable features agrees with onset of distributed nature as seen in Fig. 2

---

[5]One could have expected these results in view of the work on signal propagation Schoenholz et al. (2016) which argues that for properly initialized (random) networks similar inputs have similar representations

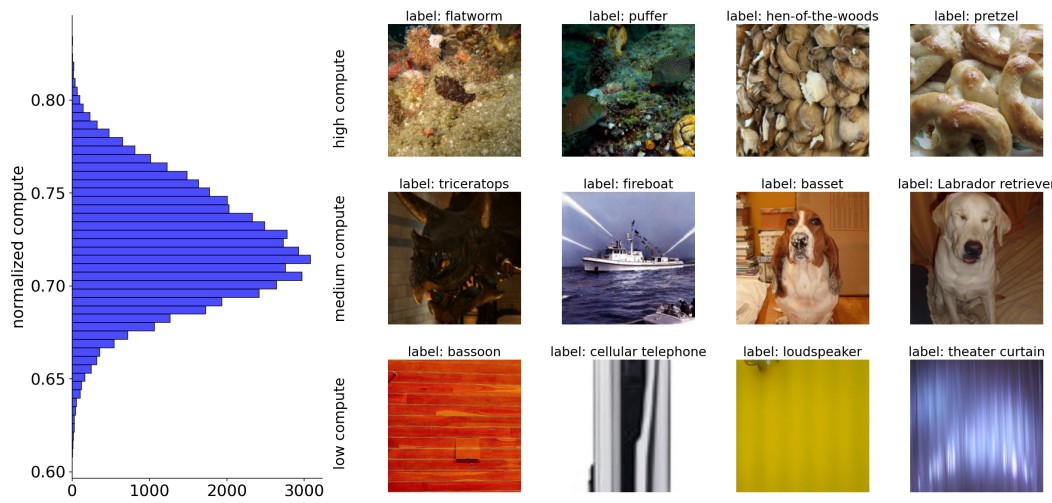

Figure 5: Distribution of computational cost on the ImageNet validation set for the top-2 DNA (25% skip) model. We observe that (i) the compute follows roughly Gaussian distribution; we believe that it is a reflection of the dataset rather than the model. (ii) the amount of compute that the model spends on the image correlates with the visual content. Namely, the model prioritizes boundary patches, so in high compute images almost everything looks like a boundary, whereas in low compute images almost everything is a background.

### 3.3 EFFICIENCY

**Compute.** We find that the model allocates compute in an interpretable manner, suggesting that its decisions are contextual. We examine this behavior using the top-2 DNA model (with 25% skip rate) shown in Fig. 2 by measuring the compute allocated to each image. Specifically, we count the number of computational modules used along the ribbon of each patch, average this count over all patches within an image, and normalize the result such that the normalized compute equals 1 when the ribbon never skips a module. The results are shown in Fig. 5. On the left, we plot the distribution of compute across all images in the ImageNet validation set. On the right, we randomly select four images for each compute level—high (top 1% of images), medium (49%–51%), and low (bottom 1%). We observe that images requiring lower compute tend to be visually simpler, containing no "object", while the high compute images are vibrant with textures and the model needs more compute to segment it.

**Parameter sharing.** We observe that both top-1 and top-2 models tend to reuse certain modules, leading to *emergent*, input-dependent weight sharing. The models have not been explicitly incentivized to share parameters. On Fig. 7 we show several results: First, weight-sharing distribution is roughly gaussian with 25% and 15% of parameters reused. Second, high-reuse images do not contain the object. The models like to reuse the module that has full (*i.e.* not sparse) attention. We also verified that images that use the least active parameters are classified correctly. We find that emergent parameter sharing developed by the models relies on similar features. We can conclude this because different models exhibit similar amount of parameter sharing on the same images. Finally, we find that compute savings and parameter savings are not correlated. See Appendix C for details.

## 4 LANGUAGE DNA MODELS

In this Section, we discuss DNA language models. Before diving in, we emphasize that language and image datasets (and objectives) are *very* different. In particular, FineWeb-edu is vastly more complex than ImageNet. Consequently, our models are way too small to truly absorb it (or even 21B token part of it). Nonetheless, already at this small scale and vastly "underparametrized" regime we observe many interesting effects.

## 4.1 EXPERIMENTAL DETAILS

We trained several DNA models at a scale comparable to GPT-2 Medium Radford et al. (2019), with the complete set of hyperparameters listed in Table 2. All models were trained on the FineWeb-Edu Lozhkov et al. (2024) 100B-token subset, for approximately 21B tokens in total ($40,000$ steps with a batch size of $512$ and a context length of $1024$). For each model, we searched over three learning rates, $\eta \in \{0.0004, 0.0008, 0.0016\}$, with a fixed weight decay value $\lambda = 0.1$. All models are trained using the same learning rate schedule, which includes a linear warmup for the first $1,000$ steps and a linear decay to $0.1 \cdot \eta$ for the last $8,000$ steps. The training curves of the best run for each model, together with the routing patterns of the top-1 and top-2 DNA models, are shown in Fig. 6. We present a few standard benchmark results in Table 3.

| Model | $N_b$ | $N_m$ | $N_r$ | $d_{\text{embed}}$ | $d_{\text{MLP}}$ | $N_{\text{head}}$ | Active Params | Params |
|---|---|---|---|---|---|---|---|---|
| GPT-2 medium | - | 24 | - | 1024 | 4096 | 16 | 406M | 406M |
| top-1 DNA | 2 | 36 | 22 | 1024 | 4096 | 16 | 406M (242M) | 583M |
| top-2 DNA | 2 | 72 | 24 | 768 | 3072 | 12 | 433M (266M) | 603M |

Table 2: Hyperparameters of DNA models and GPT-2 medium (no weight-tying). Note that for DNA models, the number of active parameters shown in parentheses refers to non-shared active parameters (as detailed in Sec. 4.3).

## 4.2 INTERPRETABILITY

**Early routers group similar tokens.** We selected two example paragraphs: One from Wikipedia and one hand-crafted (see Fig. H.1 for the full paragraphs and Fig. 18,19 for full token flows). Then we visualized the routing decisions made by $R_1$ in the left panel of Fig. 8. Each token is colored according to the CDF value of the path it follows, using the same color scheme as in Fig. 1(d, f). We found that the router $R_1$ consistently sends semantically similar words to $M_9$, punctuation marks to $M_{27}$, and word pieces to $M_{29}$ in both examples. In the Wikipedia example, the router additionally groups plural nouns and routes them to $M_1$, verb variants to $M_{10}$, and related prepositions to $M_{25}$ and $M_{30}$ accordingly.

| Model | Loss ($\downarrow$) | ARC-E ($\uparrow$) | BoolQ ($\uparrow$) | HellaS ($\uparrow$) | LAMBADA ($\uparrow$) | PIQA ($\uparrow$) | RACE ($\uparrow$) | Wiki ($\downarrow$) |
|---|---|---|---|---|---|---|---|---|
| Random | 10.825 | 25.0 | 50.0 | 25.0 | 0.0 | 50.0 | 25.0 | $\sim$ inf |
| GPT-2 (406M) | 2.720 | 58.9 | 60.5 | 40.5 | 33.8 | 66.9 | **32.3** | 33.7 |
| top-1 (406M) | 2.754 | 56.9 | 60.8 | 38.6 | 28.7 | 65.8 | 30.9 | 38.7 |
| top-2 (433M) | **2.674** | **59.2** | **61.0** | **41.8** | **34.0** | **67.9** | 31.1 | **31.5** |
| GPT-2 (30% shallower) | 2.772 | 58.0 | 54.9 | 37.9 | 31.4 | 65.9 | 30.1 | 38.0 |
| top-2 (30% skip) | 2.784 | 52.5 | 52.9 | 35.5 | 23.8 | 64.2 | 28.1 | 52.6 |

Table 3: Final validation loss and zero-shot downstream evaluation. We reported accuracy in all cases, except for Wikitext, where we reported word-level perplexity. Hyperparameter details for the model with skipping and the shallower GPT-2 are listed in Appendix A.

**Interpreting path distribution.** We start with the NLP version of Fig. 3. Namely, we visualize the tokens that follow the same path for several different ranks in the right panel of Fig. 8. As expected, low-rank (*i.e.* frequent) paths tend to capture frequently used words, such as linking verbs, or words that share similar conceptual roles (e.g., rank-8 corresponds to relationships between actions and their targets or contexts; rank-64 relates to human endeavors). We also find that rank-2 path focuses on end of sentence tokens, which can be interpreted as sentence-level attention. In contrast, mid- and high-rank (*i.e.* rare) paths exhibit more variability. These include, for example, distinct adverbs (rank-16384), or even commonly used words appearing in unexpectedly high-rank paths (e.g., ranks 1024, 32768, 65536, etc.). This may seem surprising, as one might expect high-rank paths to capture rare or domain-specific concepts. However, we hypothesize that due to the highly contextual nature of language, common words routed through high-rank paths likely carry context-specific information rather than simply reflecting their base lexical identity. A more detailed investigation of this phenomenon at a larger scale is left for future work.

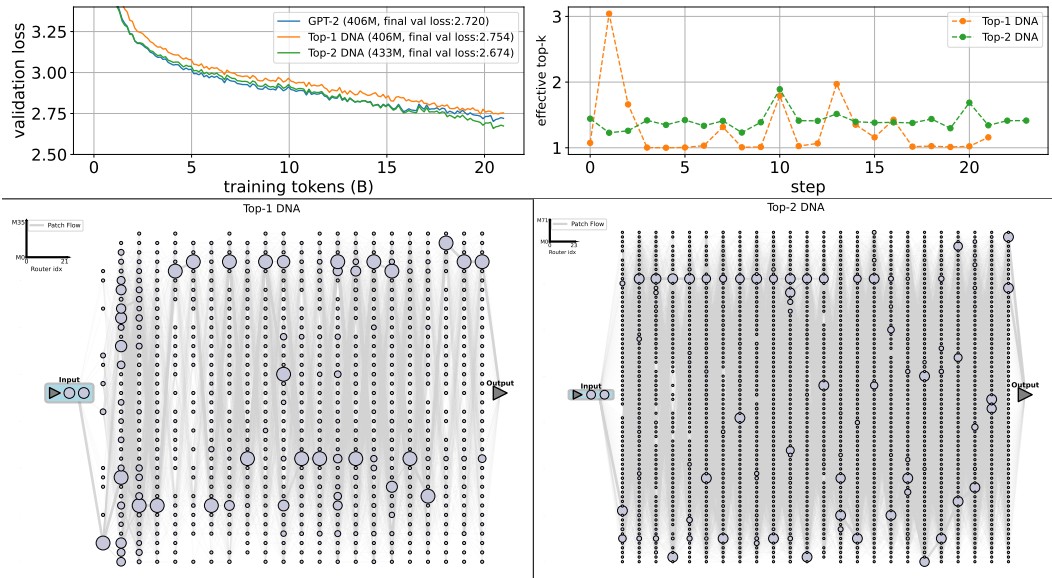

Figure 6: Models trained with 21B tokens sampled from FineWeb-Edu dataset. **Top-Left**: Validation loss measured on a subset that we leave aside. **Top-Right**: Effective number of compute nodes used per step for DNA models at the end of training. The two models exhibit different behaviors in the early steps. Compared to the patterns observed in vision models, they both follow different trends as the number of steps increases. **Bottom**: Flow of tokens plotted using the validation set of the FineWeb-Edu dataset for the top-1 (left) and top-2 (right) DNA models.

## 4.3 EFFICIENCY

**Compute.** Similar to the vision case, we trained a top-2 DNA model that targets skipping $30\%$ of the tokens, where most hyperparameters are the same as those shown in Fig. 6, except $N_b = 1$ and $N_r$=23 in this case. We find that (i) compute distribution for images and text is very different showing that text has significantly higher diversity/complexity (see Fig. 5) leading to majority of examples requiring (roughly) the same amount of compute, (ii) There are documents that require much lower compute. By inspection we find that these documents are qualitatively different from the "average examples" in that they either contain HTML code, large number of links, are parts of bibliography, contain many characters from the languages that our model has not learned yet such as Arabic, Greek, Hebrew etc.

**Parameter sharing.** We find that language DNA models also tend to like parameter sharing. However, unlike in the vision case, we find that parameters sharing does not correlate with any notable features in the text. We also make a more careful check by measuring two types of correlation (similar to the vision case). First, we check if module reuse is correlated to the module skip (Fig. 11 (d) in Appendix D). Unlike in the vision case, we find a strong correlation. However, we believe that this correlation has a simple explanation: if fewer modules are skipped, the probability of reuse goes up. Second we check if there are any similarities between text documents that yield higher module reuse. Unlike in the vision case, we find that there is no correlation between two different DNA models. Consequently, we conclude that module reuse is most likely random in the language case. This suggests that language DNAs can be further improved by discouraging module reuse.

## 5 CONCLUSION

In this work we have introduced distributed neural architectures that process each token differently, depending on its content and context. We introduced initial model architectures and showed that they are trainable and competitive with the corresponding dense baselines in both vision and language domains. We also showed that DNA models can be trained to allocate compute intelligently based on the input, and this allocation is interpretable.

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

## A    EXPERIMENTAL DETAILS

In this section, we list all the details that the reader might find helpful.

**Hardware and Precision** All experiments were conducted using a codebase based on the PyTorch (Paszke et al., 2019) and xFormers (Lefaudeux et al., 2022) libraries. Training was performed in mixed precision using BFloat16 on NVIDIA A100 80GB SXM4 GPUs and NVIDIA H200 SXM5 GPUs.

**Memory/Throughput** In all cases, our implementation runs slower and consumes more memory compared to its dense counterpart. This overhead arises primarily from the unoptimized handling of dynamically changing sequence lengths across batches and internal steps/routers when multiple modules are activated. Due to the highly dynamic nature of the model, it is infeasible to precompute and cache attention masks for all possible sequence length combinations. As a result, we currently generate attention masks on the fly at each step and for each module.

We also find that increasing the batch size can improve GPU utilization. However, this comes with trade-offs: for the vision model, it degrades performance, while for the language model, memory becomes the limiting factor. Part of this memory bottleneck stems from the worst-case scenario, where the model must store intermediate results for all modules across all steps and routers.

Nevertheless, we do not view this as a fundamental limitation. In our long-term vision of a fully distributed model, module locality will be introduced, which can alleviate this bottleneck and decouple memory constraints from architectural design. A thorough investigation of this direction is left for future work.

**Effective top-$k$** To measure the diversity of routers' decisions, we introduce the concept of *effective top-$k$*, defined as the inverse of $\text{IPR}_\alpha^{(s)}$ for a given router at step $s$.

The quantity $\text{IPR}_\alpha^{(s)}$ is computed from the token counts $c_i^{(s)}$ as follows:

$$\text{IPR}_\alpha^{(s)} = \frac{\sum_{i \in \star} (c_i^{(s)})^{2\alpha}}{\left(\sum_{i \in \star} (c_i^{(s)})^2\right)^\alpha} . \tag{4}$$

By definition, $\text{IPR}_\alpha^{(s)}$ lies in the range $[N_m^{-\alpha+1}, 1]$. It attains its maximum value when only one $c_i^{(s)}$ is non-zero, and its minimum when all $c^{(s)}i$ are equal at step $s$. The exponent $\alpha$ controls the sensitivity of $\text{IPR}_\alpha^{(s)}$ to the distribution of the $c_i^{(s)}$ values. Throughout this work, we use $\alpha = 1.5$.

### A.1    VISION MODELS

**Architecture** For all models listed in Table 4, we use image size $224 \times 224$ with a patch size of $16 \times 16$. We are not using `[CLS]` token, but instead use global average pooling before the final output.

**Initialization** We initialize all weights but the patch embedding using PyTorch truncated normal with 0 mean and 0.02 standard deviation, where the patch embedding was implemented with `nn.Conv2d` with the default initialization. No learnable biases used in any module.

| Model | $N_b$ | $N_m$ | $N_r$ | $d_{\text{embed}}$ | $d_{\text{MLP}}$ | $N_{\text{head}}$ | Active Params | Params |
|---|---|---|---|---|---|---|---|---|
| ViT-small | - | 12 | - | 384 | 1536 | 6 | 22M | 22M |
| top-1 DNA | 1 | 18 | 11 | 384 | 1536 | 6 | 22M | 34M |
| top-2 DNA (25% skip) | 1 | 24 | 11 | 256 | 1024 | 4 | 18M | 18M |
| top-1 DNA-n | 1 | 18(Attn) + 18(MLP) | 22 | 384 | 1536 | 6 | 18M-26M | 34M |
| top-2 DNA-n | 1 | 24(Attn) + 24(MLP) | 22 | 384 | 1536 | 6 | 38M-40M | 44M |

Table 4: Hyperparameters of all vision models used in this paper, where only the last two lines are new compared to Fig. 1. Note that i) the number of active parameters column does not take model re-use into account; ii) the backbone for DNA-n models are full transformer blocks.

**Optimization** We use AdamW optimizer with $\beta_1 = 0.9$ and $\beta_2 = 0.99$, $\epsilon = 1 \times 10^{-8}$. All models were trained for 300 epochs with a 10-epoch warmup, following a cos-decay. The initial learning rate is set to $\eta_{\text{init}} = 1 \times 10^{-7}$ and final learning rate is set to $\eta_{\text{final}} = 1 \times 10^{-6}$.

**Augmentation** We use PyTorch's default choices for hyperparameters in data augmentations. For MixUp, we set $\alpha = 0.8$. We randomly select either MixUp or CutMix with equal probability for each batch. We apply ColorJitter with parameters: brightness=0.3, contrast=0.3, and saturation=0.3.

### A.2 LANGUAGE MODELS

**Architecture** We use models with dimension of each head, i.e. $d_{\text{embed}}/N_{\text{head}}$ fixed as 64. All other details are listed in Table 5.

| Model | $N_b$ | $N_m$ | $N_r$ | $d_{\text{embed}}$ | $d_{\text{MLP}}$ | $N_{\text{head}}$ | Active Params | Params |
|---|---|---|---|---|---|---|---|---|
| GPT-2 medium | - | 24 | - | 1024 | 4096 | 16 | 406M | 406M |
| top-1 DNA | 2 | 36 | 22 | 1024 | 4096 | 16 | 406M | 583M |
| top-2 DNA | 2 | 72 | 24 | 768 | 3072 | 12 | 433M | 603M |
| top-1 DNA-n | 2 | 36(Attn) + 36(MLP) | 44 | 1024 | 4096 | 16 | $347 - 465$M | 583M |
| top-2 DNA-n | 2 | 64(Attn) + 64(MLP) | 44 | 768 | 3072 | 12 | $350 - 455$M | 546M |
| GPT-2 (30% shallower) | - | 17 | - | 1024 | 4096 | 16 | 313M | 313M |
| top-2 DNA (30% skip) | 1 | 72 | 23 | 768 | 3072 | 12 | 412M | 596M |

Table 5: Hyperparameters of DNA models and GPT-2 medium (no weight-tying). Note that for DNA models, we do not take module re-use into account while counting the number of active parameters.

**Tokenizer** We use the GPT-2 Tokenizer from tiktoken library. The vocabulary size is $50,257$.

**Initialization** We initialize all weights using PyTorch truncated normal with 0 mean and 0.02 standard deviation. No learnable biases used in any module.

**Optimization** We use AdamW optimizer with $\beta_1 = 0.9$ and $\beta_2 = 0.95$, $\epsilon = 1 \times 10^{-15}$. All models were trained for $40,000$ steps with a $1,000$-step warmup, following a constant period then linear decay to $0.1\eta$ for last 20% of training steps. The initial learning rate is set to $\eta_{\text{init}} = 1 \times 10^{-7}$ and final learning rate is set to $\eta_{\text{final}} = 1 \times 10^{-6}$.

**Downstream Evaluation** Downstream Evaluations were done using torchtune recipes (torchtune maintainers & contributors, 2024) and lm-evaluation-harness library (Gao et al., 2023). The dataset we used for evaluation are listed here: Arc-Easy (Clark et al., 2018), BoolQ (Clark et al., 2019), HellaSwag (Zellers et al., 2019), LAMBADA OpenAI version (Radford et al., 2019), PIQA (Bisk et al., 2020), RACE (Lai et al., 2017) and Wikitext-2 (Merity et al., 2016).

## B RELATION TO OTHER WORK

**DNAs as a generalization of Mixture-of-X.** The simplest way to arrive to DNAs is to imagine that we are trying to implement MoE and MoD at the same time. In that case, each token skips MLP blocks as well as certain transformer blocks leading to different tokens following paths of different lengths. We assume that there is no inherent reason for (i) all-to-all attention in *each* layer, (ii) ordering modules by depth and (iii) attaching MLP layers to each attention operation. Relaxing (i)-(iii) and allowing routers to act on individual tokens to retain causality leads to DNAs.

**DNAs as soft neural architecture search.** Neural architecture search (NAS) is a program that attempts using optimization (*i.e.* RL) to learn the most performant composition of modules Zoph & Le (2016). NAS is computationally prohibitive because it requires a large number of full training runs. There does exist a differentiable version of NAS (known as DARTS) Liu et al. (2018), however it is conceptually different from DNA. Indeed, after the architecture search is over, NAS is a static architecture that processes all inputs through the same computational pathways. On the other hand, DNA determines these pathways based on the input[6]. Different tokens, in fact, see *different* architecture, with different number of active parameters and different amount of compute. What's common is that this (data-dependent) connectivity emerges as a result of end-to-end learning. So the reader can view DNA as a very soft form of NAS.

**Relation to ensemble methods.** Another productive way to understand DNAs is from the perspective of ensembles Hansen & Salamon (1990). The most direct way of ensembling involves training several neural networks and then sampling them to establish consensus. This technique is very powerful, however it dramatically increases both training and inference costs. An ingenious way to emulate an ensemble is known as *dropout* Srivastava et al. (2014). It approximates training an ensemble of networks by (randomly) sampling small sub-networks for each input and optimizing them in parallel. At inference time we call the entire network with scaled weights approximating the ensemble-average[7].

Routing can be viewed as a generalization of ensembling, where the sub-network choice is not random, but depends on data. Indeed, at training time each token (or datapoint) chooses a sub-network based on a collection of routing decisions. During optimization different sub-networks adapt to different inputs. At inference time the most likely sub-network is used for each token. Both MoE and DNA can be thought of as such "smart" ensembles. In fact, the ribbon introduced above is a compact way of specifying the sub-network activated for each token. An interesting early example of dynamic routing is Sabour et al. (2017). Another way to contrast this to dropout-like methods is to observe from Fig. 1 (c),(d) that frequency of using the sub-networks follows a power-law, whereas in stochastic case it would be uniform. We find that these sub-networks are highly interpretable units in routed architectures.

**Dynamically sparse attention.** Routing is often applied to the MLP modules. Since MLP performs computation on individual tokens the implementation of routing is straight-forward. However, the routing of attention mechanism requires some explanation. From the perspective of the attention modules, routing can be viewed as dynamic sparsity. As explained in Fig. 1, the routing is (i) fully causal and (ii) at each step of the forward pass, each attention module computes the attention matrix *only* over the tokens it has received. The tokens that were skipped at a given step do not participate in any attention at that step. This operation can be interpreted as (slightly generalized, data-dependent) sparse attention operation. The attention pattern is often human-interpretable as can be seen in Fig. 13 and Fig. 8.

**Interpretability of trained DNA models.** Interpretability research traditionally focuses on the attention maps, activation patterns in the MLP layers, or (in older computer vision settings) on channels in the convolutional layers. Routed models (including MoE, MoD and DNA) present a new path for interpretability research because there is a routing mechanism that can be analyzed independently of activations. Each token follows a path through the network. Each such path is assembled from a collection of routing decisions. Consequently, it is natural to attempt to interpret these paths as well as the individual routing decisions. For example, we find that maximizing the probability that an image follows a particular path allows us to reconstruct the essential features of the image Fig. 4. We perform a collection of experiments interpreting routing decisions, ultimately confirming that routers contain human-understandable structure.

---

[6]DARTS can be generalized to resemble DNA more if we allow the module choice to be data-dependent. To be precise, the variables $\alpha_o^{(i,j)}$ should depend on input $x$.

[7]For deeper discussion of ensemble-averaging interpretation of dropout see Gal & Ghahramani (2016), where it is related to uncertainty quantification.

## C    FURTHER VISUALIZATIONS FOR VISION DNAS

Fig. 7 illustrates the distribution of parameter-sharing in top-1 and top-2 vision DNAs. We find that parameter-sharing emerges without explicit nudging. The images characterized by high parameter sharing are complex, but without an obvious object/background separation - scenery and images heavy on text.

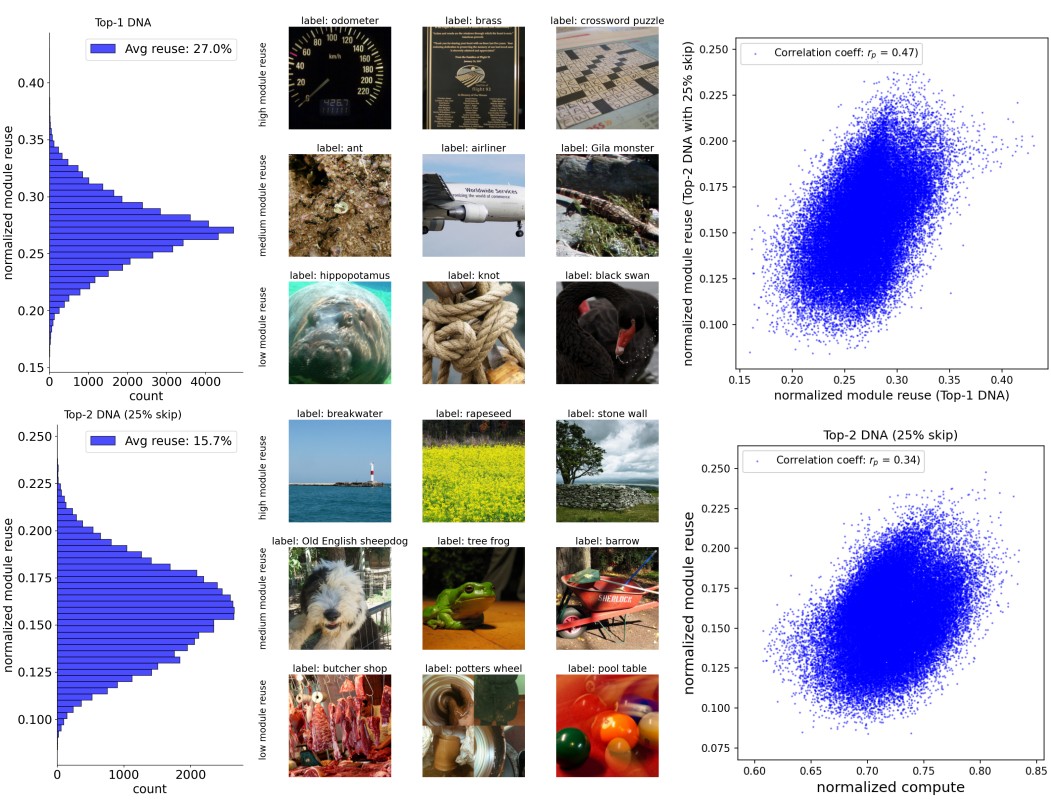

Figure 7: **Left**. Statistics of module reuse and representative images for the top-1 and top-2 DNA models. We see that high model reuse is dedicated to the complex images that lack an object to be segmented. For those images attention tends to spread everywhere. Low module reuse images are less interpretable, but we surmise that most of the image is taken by the object. **Top-Right** Correlation between module reuse for top-1 and top-2 DNA models. Both models tend to reuse modules on the same images (high correlation). **Bottom-Right**. Correlation between module skip and module reuse for the top-2 DNA model. The model uses different features to save compute and to reuse modules (low correlation).

## D    FURTHER VISUALIZATIONS FOR LANGUAGE DNAS

In this section, we provide further visualizations for language DNAs. In Fig. 8, we show the type of tokens that are accepted by different nodes. We also show examples of tokens that follow different paths.

Fig. 9 shows token flows through the DNA with compute penalty. We see behavior similar to vision DNA where dense network turns into a distributed one.

Fig. 10 shows the compute distribution of different documents from fineweb-edu. We find that some documents do take small amount of compute. These documents tend to be qualitatively less informative for the model: either containing foreign languages or html links. Fig 7 shows the patterns of parameter reuse. Unlike the vision case, we find no discernible patterns for model parameter re-use.

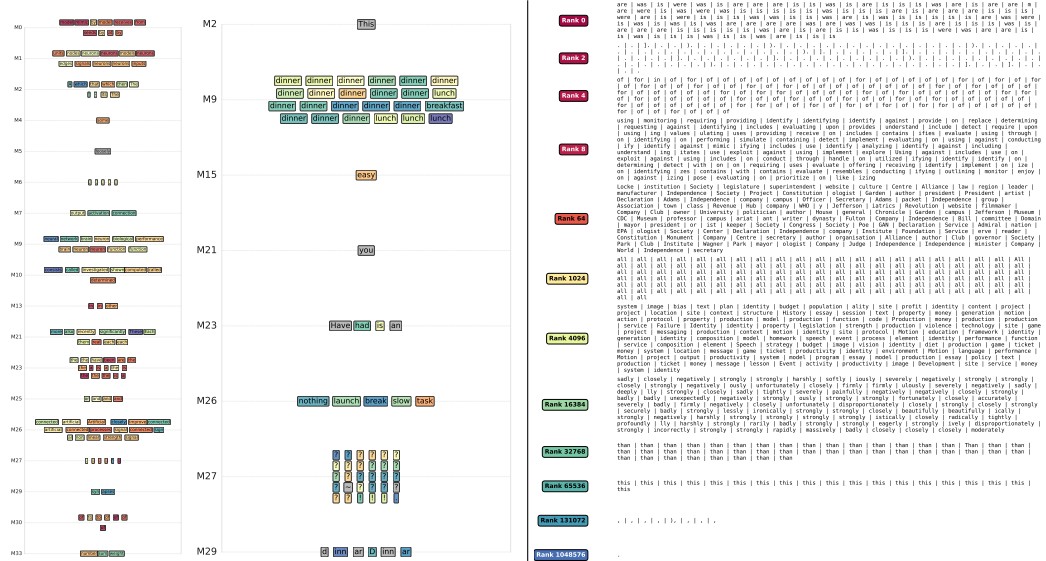

Figure 8: **Left**: Two examples illustrate how the router $R_1$ (at step-1) directs semantically/lexically similar tokens to specialized modules. For instance, $M_9$ groups tokens related to neural concepts in the first example and to "breakfast, lunch, dinner" in the second, whereas $M_{29}$ tends to receive word-pieces that can be combined into whole words. **Right**: Tokens (separated by |) traverse distinct paths that align with semantic/lexical categories.

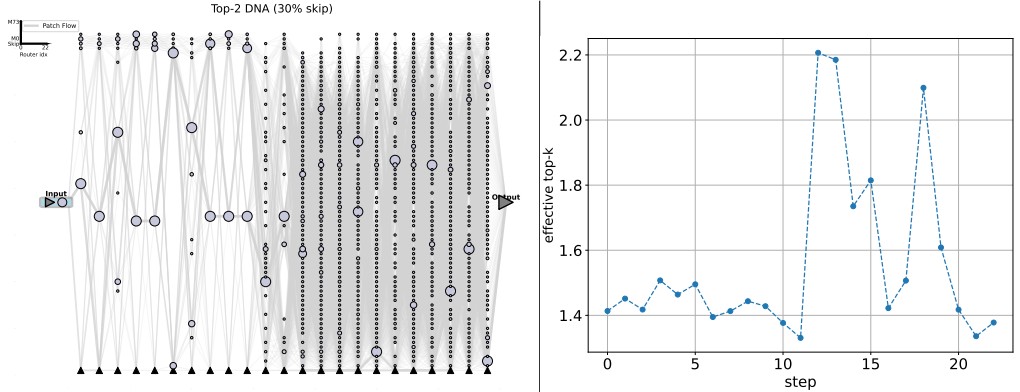

Figure 9: top-2 DNA model with a $30\%$ skip rate (final validation loss 2.784). **Left**: Token flow on the FineWeb-Edu validation set. Unlike the model in Fig. 6, this model makes similar routing decisions across most tokens in the first few steps, which is similar to the vision cases in Fig. 2. **Right**: Effective top-$k$ measured based on the token flow.

# E MODULE USAGE AND LOAD BALANCING

We plot the module usage distribution for all DNA models used in the main text in Fig. 12. We observe that, without load balancing, although certain modules bear a heavier load, completely dead modules are rare. This may be because we are not considering a setting with high sparsity.

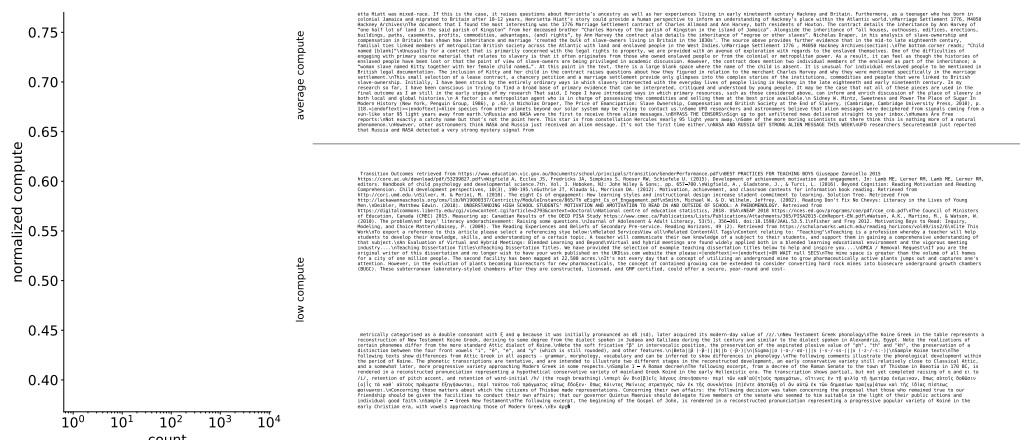

Figure 10: **Left**: Compute distribution measured on the validation set of the FineWeb-Edu dataset for the top-2 DNA model with $30\%$ skip. We observe that the distribution is very heavily peaked around *above* the compute threshold of $30\%$ (note the $\log$-scale of $x$-axis!). Nonetheless, some documents are low-compute as we show on the right. **Right**: Selected tokens that pass through ribbons with different compute. We find that sequences associated with very low-compute ribbons either contain HTML code, large number of web links, are parts of bibliography, contain many characters from the languages that our model has not learned yet such as Arabic, Greek, Hebrew etc..

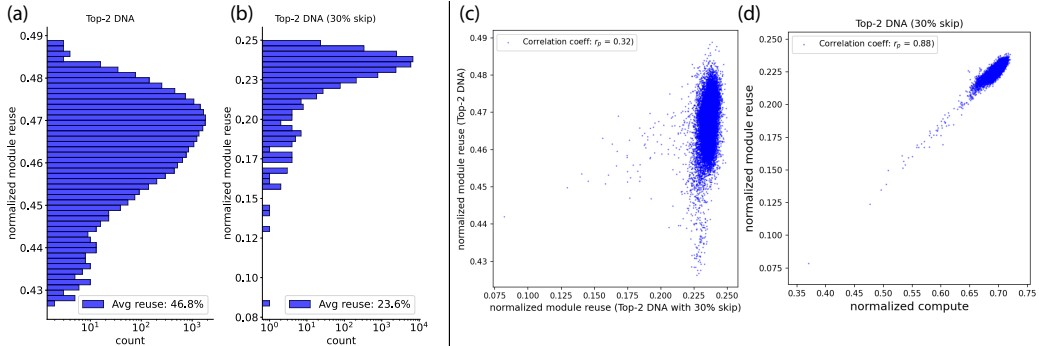

Figure 11: **(a),(b)** Statistics of module reuse and representative sequences for the top-2 DNA models with/without skipping. We find that top-2 model reuses about $50\%$ of the modules, while the model trained to skip reuses $\approx 23\%$. **(c)** Correlation between module reuse of two different top-2 DNA models (on with module skipping and one without). We find lack of correlation suggesting that skipping decisions are arbitrary. **(d)** Correlation between module skip and module reuse. We find a strong correlation, however we believe that is has a simple statistical nature: if a module is not skipped, it is more likely to be reused.

## F DREAMING VISUALIZATIONS

### F.1 SOME DETAILS ON DEEP-DREAM VISUALIZATION

**Deep-dream analysis of paths.** To understand routing decision and module specialization more generally, without relying on validation data, we use "activation maximization". Instead of maximizing the output of a neuron or channel as is typical, we maximize the weight or probability of tokens following a given path. We view the collection of paths taken by an image through the network as a collective variable that describes the configuration of the network. We then generate other typical images that follow the same paths as by starting with noise and maximizing the probability that each token follows the same path (or ribbon) as the initial image Fig. 4.

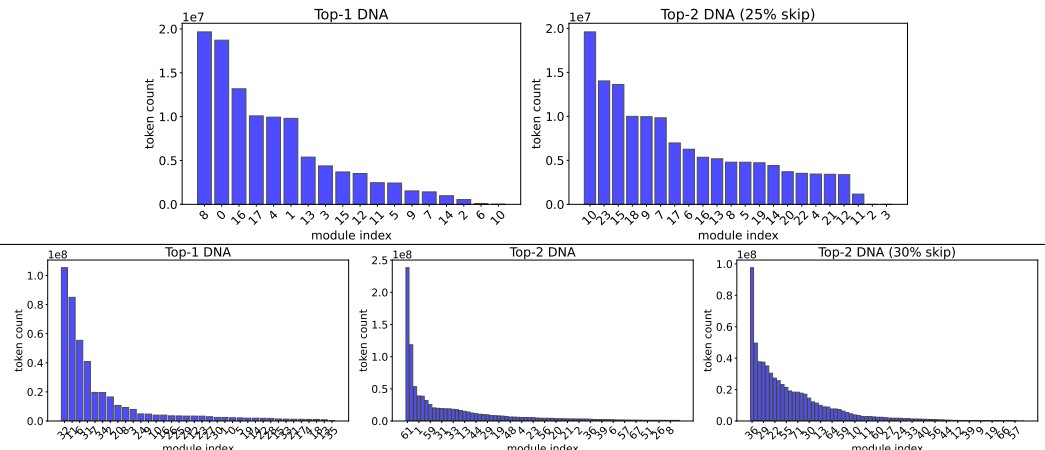

Figure 12: Patch/Token counts for each module (exclude skipping, note that the module index does not start from 0 whenever we use skip.) for all DNA models we used in main text. **Top**: vision models. **Bottom**: language models.

The simplest objective, $O_S$, is the sum of the probabilities, up to step $S \leq s_{\max}$,

$$O_S \equiv \sum_{s=1}^{S} \sum_{t} \sum_{i \in \text{Topk}_\star(\rho_\star^{(s,t)})} \rho_i^{(s,t)}, \tag{5}$$

which we maximize by performing gradient descent with respect to the input. The choices, $i$ are fixed at their original values throughout. One way to interpret this quantity is to consider $\rho$ to be the conditional probability, $\rho_i^{(s)} = \text{P}\left[i|\boldsymbol{h}^{(0)}\right]$, of the choice given the input. While $\rho$ might not be a true probability (since we sample top-$k$) or we might substitute it with the corresponding logit, the intuition will remain the same.

In this case,

$$\nabla_{\boldsymbol{h}^{(0)}} \rho_i^{(s)} = \nabla_{\boldsymbol{h}^{(0)}} \frac{\text{P}\left[\boldsymbol{h}^{(0)}|i\right] \text{P}\left[i\right]}{\text{P}\left[\boldsymbol{h}^{(0)}\right]} = \rho_i^{(s)} \nabla_{\boldsymbol{h}^{(0)}} \left(\log \text{P}\left[\boldsymbol{h}^{(0)}|i\right] - \log \text{P}\left[\boldsymbol{h}^{(0)}\right]\right), \tag{6}$$

so this procedure updates the input towards a direction which increases its probability conditioned on the choice $i$ more than decreasing the prior probability over all choices.

In Fig. 4 we see the results of optimizing the objectives $O_S$ for $S = 0, \ldots, 10$ for three selected images. It is clear from the penultimate column that optimizing $O_{10}$ results in generating a *semantically similar* image to the original input, which demonstrates that the routing decisions are highly informative about the original image. The network's choices encode information about the relative location of objects, the type of object and the separation between the objects and the background exemplified by the three reconstructed peppers, the clear reconstruction of a bird and a dog, as well as the reconstruction of a grassy background in the bird image. Looking at Eq. 6, we interpret the generated image as the one most likely to result in the particular choices made while classifying the original image. Therefore, the differences between the original and reconstructed image are also important. For example, the final reconstructed image does not maintain the colors from the original: pink peppers rather than yellow, and a white dog rather than a black one. This shows that choices are being made on the basis of specifically semantic boundaries.

Additionally, in the earlier choices, $S < 10$, the structure emerges from lower-level details building up to higher-level ones, with a sharp transition between steps 6 and 7. In the early steps, the routing choices look at edges (1-3) and on patches of light and dark (4-6). At step 7, we can suddenly see the object emerge which implies that the model begins to collect the processed information from prior steps and to classify the image. The consistency of this effect across different samples is interesting, and suggests an common structure across images. Steps 7-10 are an iterative refinement of details in

the image, such as the arrangement of animal parts and even texture[8]. Overall, we see that the router learns to intelligently route tokens through the network.

**Deep-dream analysis of segmentation.** We now use the same framework to understand why the distinct choices made by different groups of tokens (visualized in Fig. 1 through the colored overlay) are responsible for classifying the image, and how these choices rely on the context of the input image. To do this we define a more general objective which optimizes the path weights for a subset, $T$, of input tokens, and which may condition a subgroup of patches to be the same as the original input. This allows us to explore how context of the set $T$ can influence those choices made by tokens in $T$. To be concrete this objective

$$O_S^T \equiv \sum_{s=1}^{S} \sum_{t \in T} \sum_{i \in \text{Topk}_\star(\rho_\star^{(s,t)})} \rho_i^{(s,t)}, \tag{7}$$

optimizes the path weights for tokens in $T$ up to step $S$. We will choose $S = 10$ from now on.

We naturally group tokens by placing tokens that follow the same path throughout the network into the same group. After separating choices into these (disjoint) subsets we perform two experiments. First, we optimize the whole image to maximize $O_S^T$. As shown in Fig. 13 without any context this procedure reconstructs some features from the original image such as the whiteness of the dog in (d), cloth texture (e), and cloudy texture (f). Though they share some features of the original image these reconstructions lack specific detail.

We compare these reconstructions with those augmented by the original pixels of the image inside the group $T$. When we add this information, we can see that the reconstructions in the remainder of the image are much more detailed. For example, adding the boundary of the hummingbird suddenly reproduces the interior of the bird, or adding the color of the shirt reproduces the color across the shirt. This implies that context from the original image is important in explaining why certain choices were made, or what they are sensitive to in the remainder of the image.

In either case, the portion of the image reproduced is only that corresponding to the object itself (e.g. the hummingbird and not the background sky, the shirt and not the dog below, or the sky and not the hummingbird in front). This implies that the decisions, made with the context from the input, are primarily dependent on the each object separately. Based on this we hypothesize that decisions are largely related to object segmentation.

## F.2 EXPERIMENTAL DETAILS

Naive activation maximization can be done by straightforwardly maximizing any function of the internal (or external) variables of a network with respect to the input pixels. However, this tends to lead to poor results, particularly high-frequency artifacts and images without any consistent global structure. Therefore, we use several regularization methods, which serve to address these effects. These regularization methods have been drawn from several places in the literature. In particular, we follow Olah et al. (2017) and their code for the parametrization and transformations and Ghiasi et al. (2022) for the color shift and gaussian smoothing (additive noise) regularizations.

The first regularization method is to parametrize the image in Fourier space with a frequency-dependent coefficient, and to parametrize the color-channel dimension by the appropriate matrix to correct for the typical distribution of colors. Specifically, let $\theta_{\mu,c}$ be the input representation of the image in Fourier space with $\omega^\mu$ corresponding to the 2d-frequency at index $\mu$ and $c$ the color channel. We parametrize the image $\Theta$ as

$$\Theta = \text{sigmoid} \left[ \text{IFFT2} \left( \sum_c W_{\omega_x^\mu \omega_y^\mu} A_{c'c} \theta_{\mu,c} \right) \right] \tag{8}$$

with

$$A = \begin{pmatrix} 0.26 & 0.09 & 0.02 \\ 0.27 & 0.00 & -0.05 \\ 0.27 & -0.09 & 0.03 \end{pmatrix} \quad \text{and} \quad W_{\omega_\mu} = \frac{1}{\max(\omega_x^\mu, \omega_y^\mu, 224^{-1})}. \tag{9}$$

---

[8]We noticed that our models are particularly sensitive to details in images of animals and nature, which makes sense as ImageNet has a large number of these images.

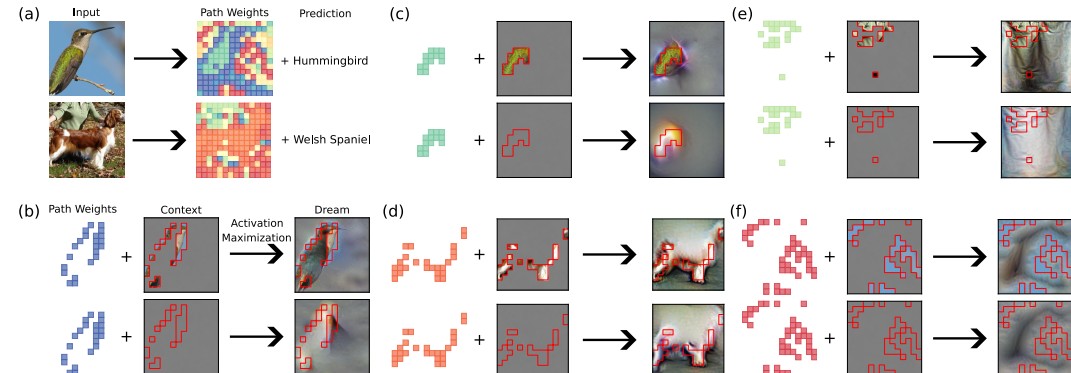

Figure 13: A schematic representation of the dreaming procedure. During inference the DNA chooses a trajectory for each token, building a computational graph towards the output shown in (a). Our dreaming procedure accepts a subset of token trajectories and context from the input and then optimizes the remainder of the input to maximize the weight on those trajectories (b-f). The red boundary delineates the group of tokens whose decisions we maximized. The upper rows have context from the original image in this region while the lower row of each example does not. The main finding is that the boundary patches have non-local information about both object and background: when these patches are provided as a context then upon dreaming procedure generates the object and some background. The patch patterns also nicely illustrate the emergent sparse attention.

Additionally we add regularization in three ways. First we randomly transform the image with the transform being re-drawn at every backward pass. The transformation is compsed of a random 6-pixel jitter, random rotation of up to 10 degrees, random scaling/shrinking by up to 10% followed by a random color shift and random per-pixel noise.

The color shift is given by the transformation $\Theta_c \to e^{\sigma_c}\Theta_c + \epsilon_c$ where $\sigma_c, \epsilon_c \sim \mathcal{N}(0, 1)$ where the noise variables depend only on the color channel and not the position in the image. The random noise is per-pixel zero-mean, and has a linearly decaying variance, starting at 1 and ending at 0 by the end of the optimization procedure.

These procedures help regularize the high-frequency components by systematically penalizing them (unless they are robust to being moved around) as well as lowering the effective learning rate on them by placing a smaller coefficient on them. Additionally, they allow global features to develop more easily because these features are more robust to noise and will therefore stabilize first, as the noise level decays through optimization.

Finally, to encourage a smoother image we explicitly add a regularizer, the total variation, which is the sum of the squares of the difference between neighboring pixels, in the horizontal, vertical, and diagonal directions. The coefficient we use for this regularization was .01.

To optimize the input image, we use the Adam optimizer with learning rate .001, and default momentum parameters. We run the optimization for 2048 steps, as mentioned with linearly decaying per-pixel noise, but with otherwise constant regularization parameters.

## F.3 FURTHER DREAMING EXAMPLES

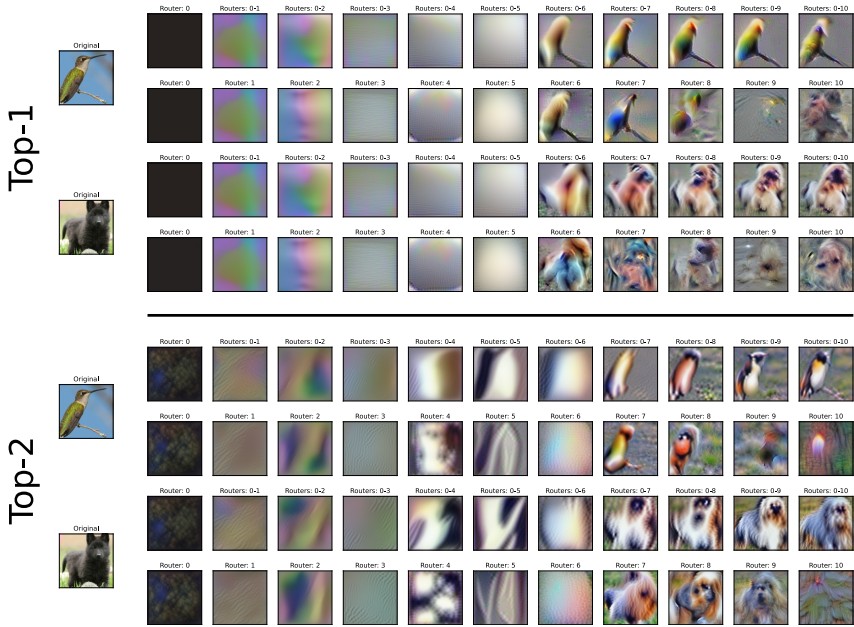

Figure 14: Reconstruction of the hummingbird and Welsh springer spaniel for the top-1 and top-2 models (some parts repeated from main text). Reconstructions are broken up into maximizing path weights from steps $0 - S$ (first and third rows) and maximizing decision weights for just step $S$ (second and fourth rows). We see that the top-2 model exhibits interesting low level features before the sixth router, while the top-1 model does not. This is because the top-1 model has an almost input-independent selection of modules before that router. Additionally, we see in both cases the same transition at router 7 where global features develop corresponding to the original image. Comparing the layer-wise and cumulative objectives, we see that the routers at the end do not look at

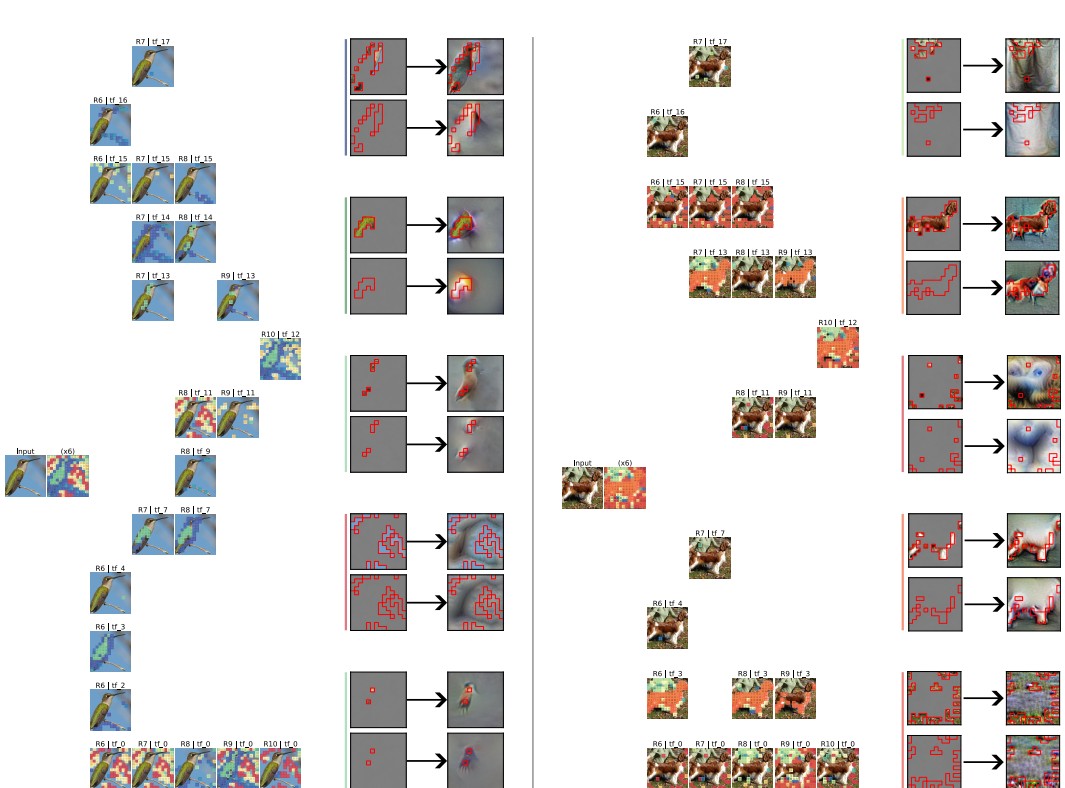

Figure 15: Series of experiments where we attempt to recover the image from the paths of a group of tokens. The red boundary delineates the group of tokens whose decisions we maximized. Each recovery comes in two varieties: The upper rows have the context from the original image in the region surrounded by the red boundary, while the lower row of each example does not. The main finding is that the boundary patches contain non-local information about both the object and the background. When these patches are provided as context, the dreaming procedure generates the object and some background. The patch patterns also nicely illustrate the emergent sparse attention.

# G MORE ON VISION DNAS

## G.1 COMPUTE DISTRIBUTION OVER PATCHES

In this subsection, we visualize the compute distribution and randomly selected patches with varying compute levels in Fig. 16.

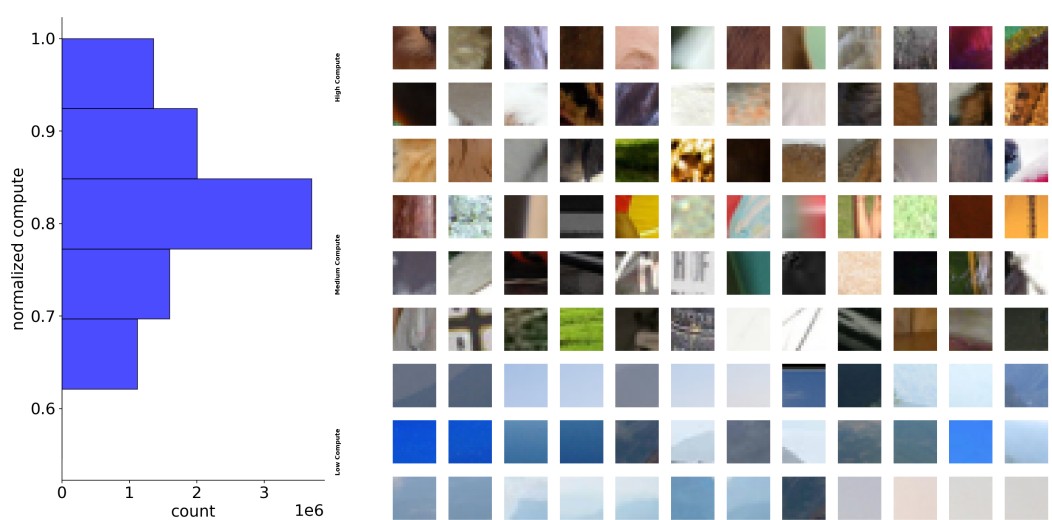

Figure 16: **Left**: Distribution of computational cost on patches from the ImageNet validation set for the top-2 DNA (25% skip) model, which is the same model as the one in Fig. 5. **Right**: High compute patches generally contain more textures and boundaries, while low compute patches tend to appear visually flatter.

## G.2 RANDOM MODEL PATHS

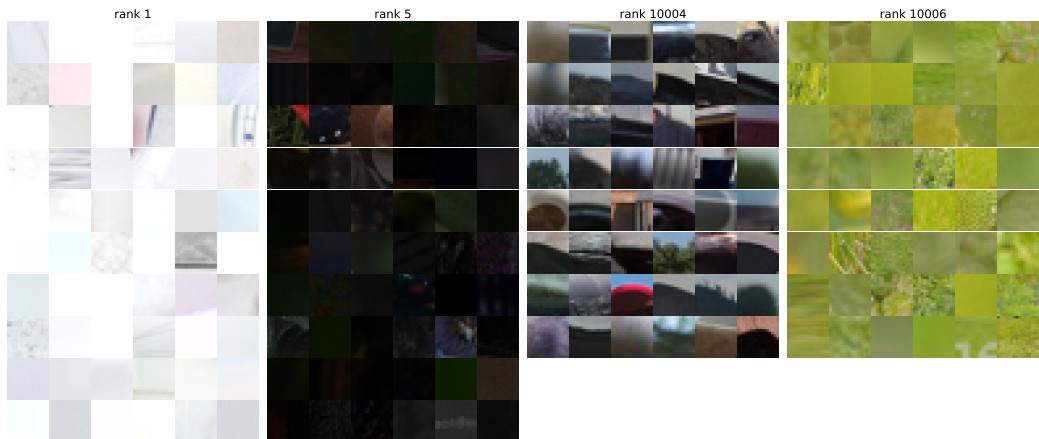

Figure 17: Illustration of randomly selected patches processed through the same ribbon in a randomly initialized top-2 DNA model. Patches following the same ribbon display strong visual similarities, likely due to the critical initialization that preserves the similarities among different input patches.

In this subsection, we demonstrate that a randomly initialized model can group patches based on only superficial similarities. The selected ribbons and corresponding patches are shown in Fig. 17. Compared to Fig. 3, we find that the patches following the same path in a randomly initialized model share much greater visual similarities. We believe this effect arises from the fact that our models

are initialized at criticality, where the correlations between different inputs are preserved as they propagate through (Schoenholz et al., 2016; Lee et al., 2018; Roberts et al., 2022).

## H  MORE ON LANGUAGE DNAS

### H.1  INTERPRETABILITY

**Examples used in main text** We used two paragraphs in the main text, where the first one is taken from `https://en.wikipedia.org/wiki/Neural_network_(machine_learning)` as it is a good test example; the second one is hand-crafted by putting in words with related concepts, together with intentional misspelling. The two examples:

> **Wiki** *A neural network consists of connected units or nodes called artificial neurons, which loosely model the neurons in the brain. Artificial neuron models that mimic biological neurons more closely have also been recently investigated and shown to significantly improve performance. These are connected by edges, which model the synapses in the brain. Each artificial neuron receives signals from connected neurons, then processes them and sends a signal to other connected neurons. The 'signal' is a real number, and the output of each neuron is computed by some non-linear function of the sum of its inputs, called the activation function. The strength of the signal at each connection is determined by a weight, which adjusts during the learning process.*

> **Crafted** *Have you had dinner? dinner? dinner? dinner? nothing? dinner? dinner? dinnar? dinner? dinner? dinner? dinner? dinner? lunch? dinner? dinner? dinner? dinner? dinner? breakfast $\sim$ dinner? dinner? dinner? Dinnar? lunch? lunch? lunch! launch! breakslow! This is an easy task.*

We present the complete token flows for these two examples using the Top-1 DNA model in Figs. 18 and 19. Note that the left panel of Fig. 8 is a zoomed-in version of those two figures.

### H.2  COMPUTE DISTRIBUTION OVER TOKENS

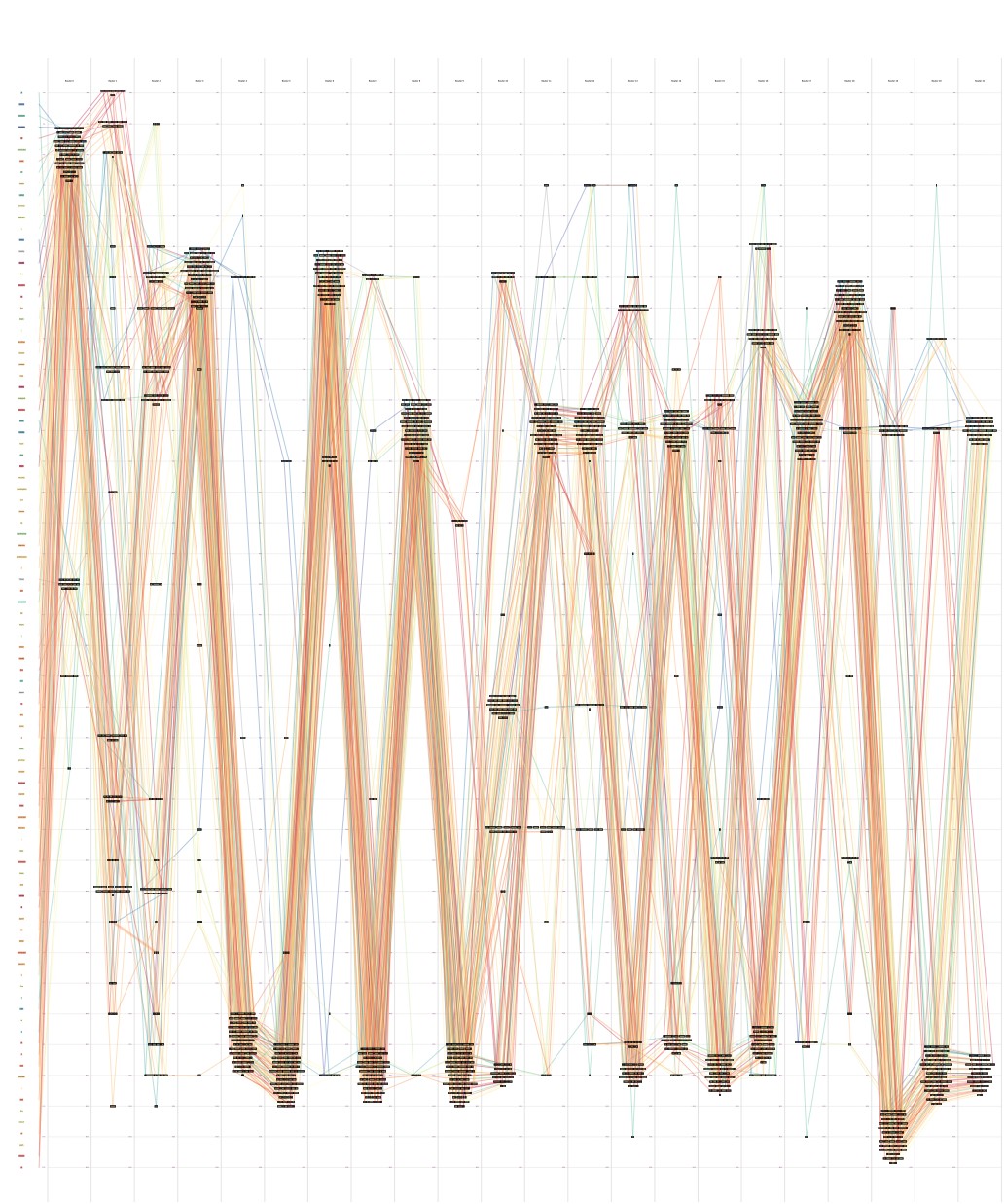

Figure 18: Full trajectories through the top-1 DNA language model on Wiki example.

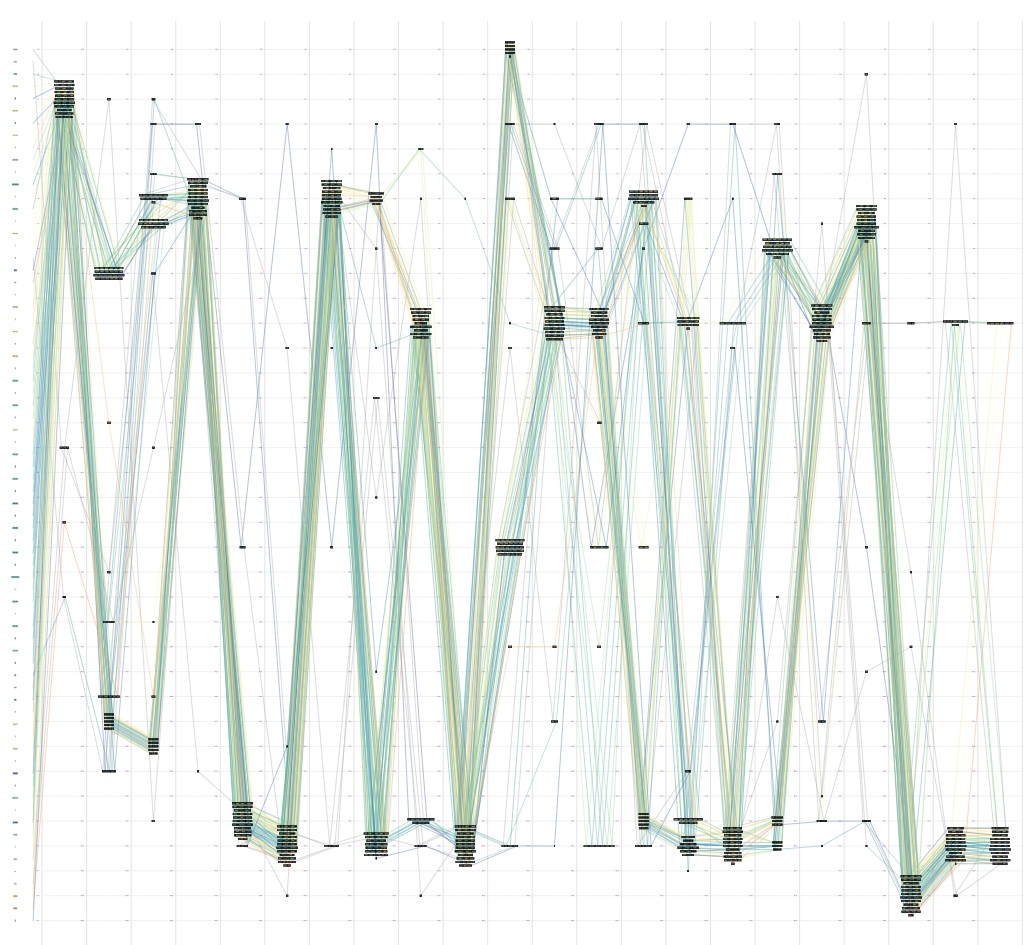

Figure 19: Full trajectories through the top-1 DNA language model on artificial example.

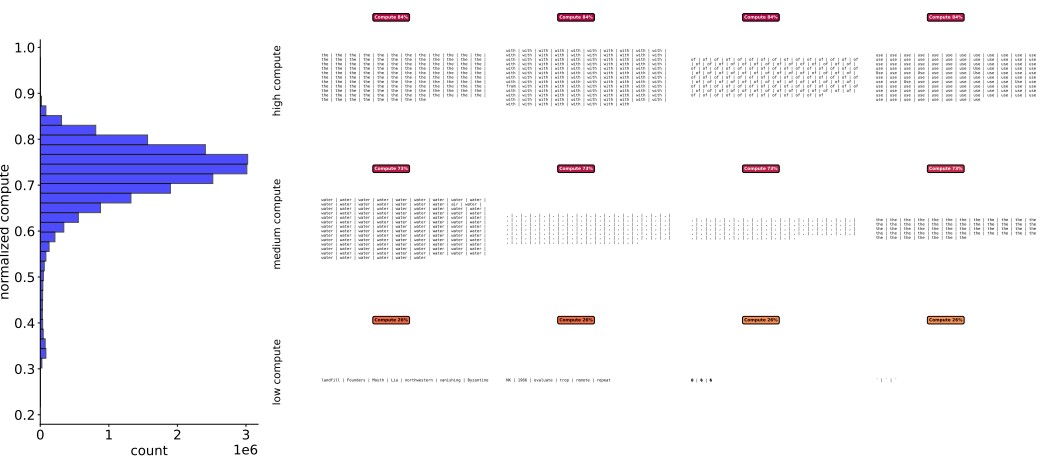

Figure 20: **Left**: Compute distribution per token measured on the validation set of the FineWeb-Edu dataset for the top-2 DNA model with 30% skip. **Right**: Selected tokens that pass through ribbons with different compute. We find that tokens associated with very low-compute ribbons are usually particular objects, concepts, or symbols that do not play a crucial role in sequences.

# I BREAKING TRANSFORMER BLOCKS

In this section, we delve deeper into architectural choices by breaking down Transformer blocks into their sub-blocks, specifically Pre-LN Attention blocks and Pre-LN MLP blocks.

## I.1 VISION DNA

We present two DNA-n models (one top-1 and one top-2) that were trained on ImageNet. Most conclusions from the main text about paths and their interpretability still hold. However, we find that in both cases of $k = 1, 2$, the models tend to utilize more Attention blocks during the initial steps and more MLP blocks closer to the output. The patch flow patterns and path/ribbon distributions for the top-1 model are shown in Fig. 21, with selected images and patches illustrating path specialization in Fig. ?? and 22. Similar figures for the top-2 model are provided in Fig. 23 and Fig. 24, respectively. We also computed effective top-$k$ at each step for both cases, as shown in Fig. 25.

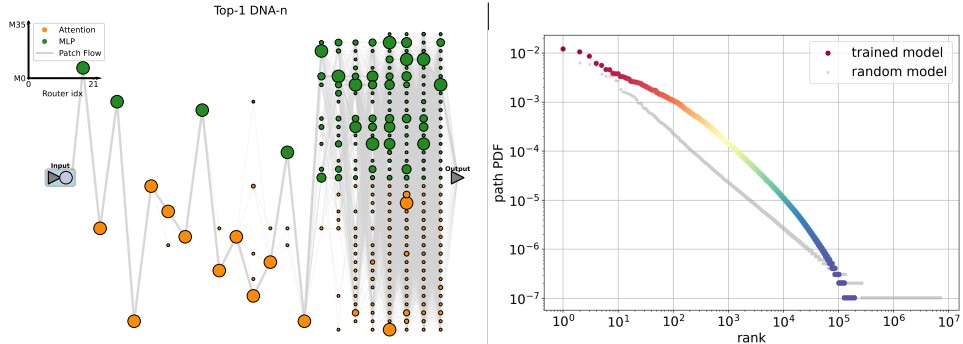

Figure 21: A top-1 DNA-n model (76.4% validation accuracy), where the Transformer blocks are separated into Attention/MLP sub-blocks. **Left** Patch flows collected over ImageNet validation set; **Right** Path distribution.

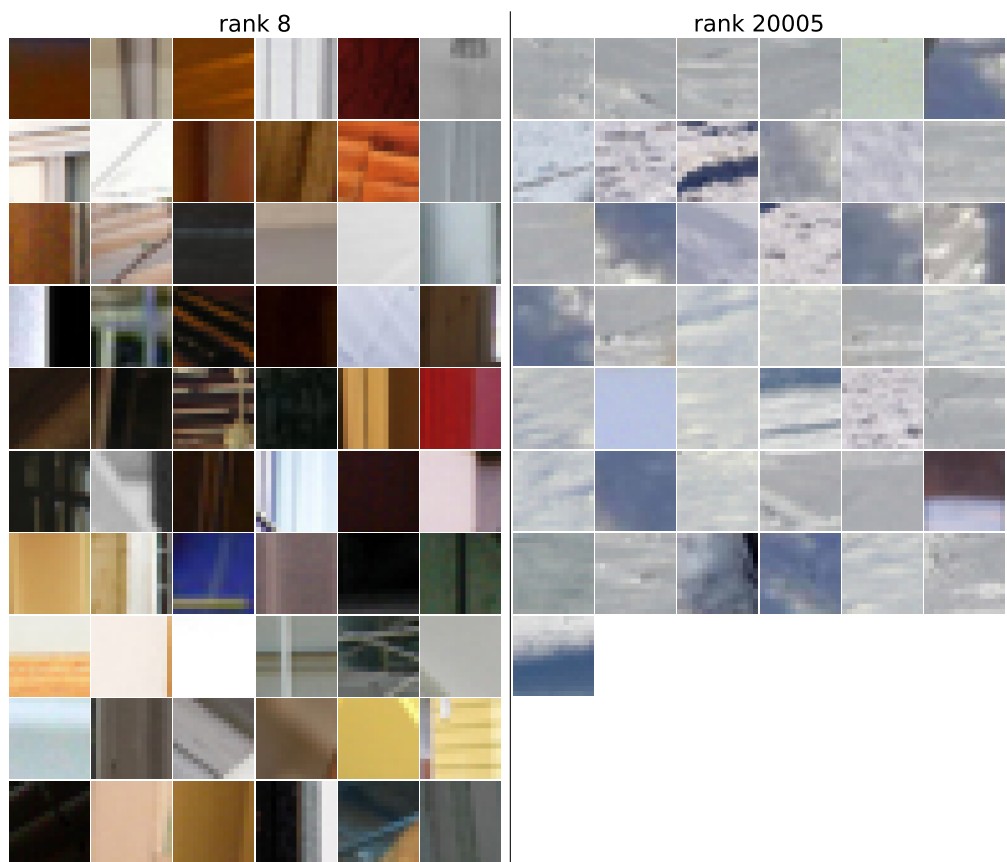

Figure 22: Visualization patches that go through the same paths for the model shown in Fig. 21. **Left**: A layout of single image goes through the model; **Left**: Patches that go through a low-rank path. **Right**: Patches that go through a high-rank path.

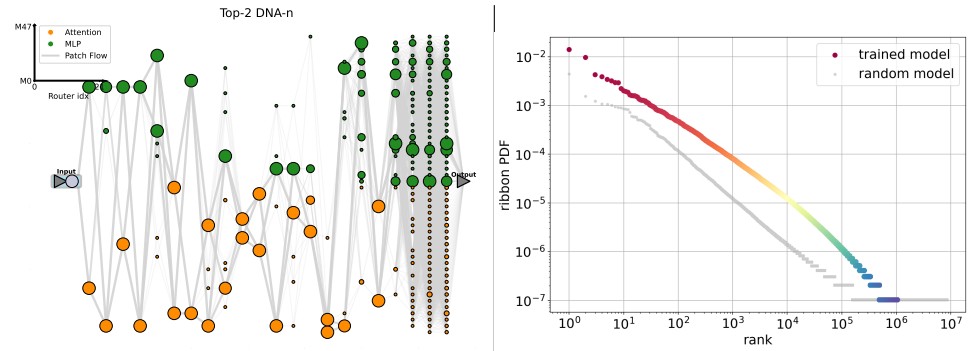

Figure 23: A top-2 DNA-n model (78.9% validation accuracy), where the Transformer blocks are separated into Attention/MLP sub-blocks. **Left**: Patch flows collected over ImageNet validation set; **Right**: Ribbon distribution.

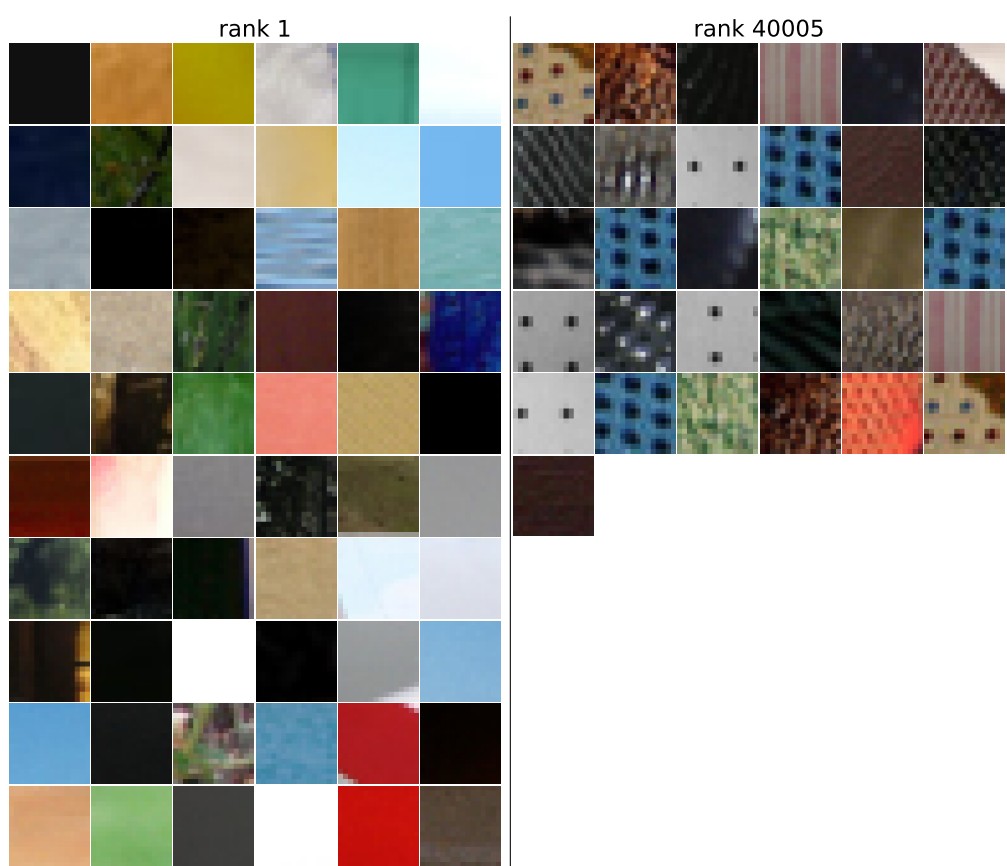

Figure 24: Visualization patch ribbons for the model shown in Fig. 23. **Left**: A layout of single image goes through the model; **Middle**: Patches that go through a low-rank ribbon; **Right**: Patches that go through a high-rank ribbon.

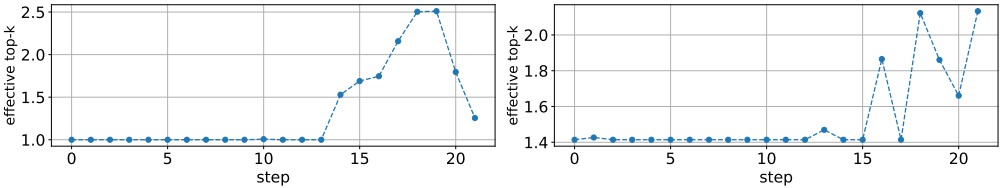

Figure 25: Effective top-$k$ for DNA-n models (vision), as a function of number of forward pass steps $s$. Early on the models are essentially dense, while later on they become more distributed. **Left**: top-1 DNA-n model. **Right**: top-2 DNA-n model.

## I.2 LANGUAGE DNA

Similarly, we trained GPT-2 Medium sized top-1 and top-2 models with separate Attention modules and MLP modules on FineWeb-Edu for 21B tokens. The token flow patterns and path/ribbon distributions are shown in Fig. 26 and 27, with the tokens that go through different paths/ribbons shown in Fig. 28. We also plot effective top-$k$ of router weights shown in Fig. 29. Similar to the separated vision models, we find that language models also tend to use more Attention modules near the input and more MLP modules closer to the output.

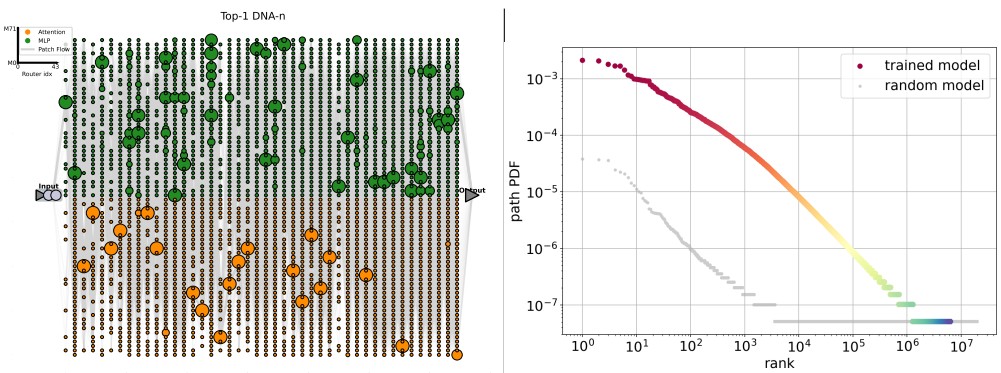

Figure 26: A top-1 DNA-n model (final validation loss 2.708), where the Transformer blocks are separated into Attention/MLP sub-blocks. **Left** Token flows collected over FineWeb-Edu validation subset; **Right** Ribbon distribution.

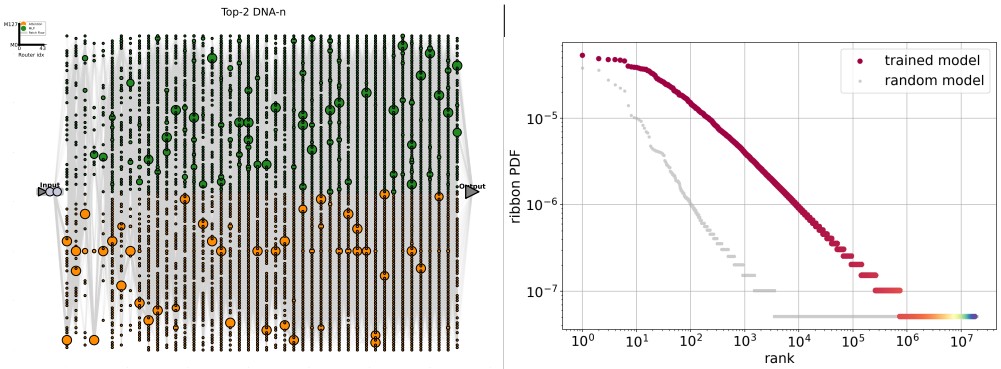

Figure 27: A top-2 DNA-n model (final validation loss 2.688), where the Transformer blocks are separated into Attention/MLP sub-blocks. **Left**: Token flows collected over FineWeb-Edu validation subset; **Right**: Ribbon distribution.

## J FUTURE WORK

We hope this work will provide a fresh take on architecture research. As we did not have either bandwidth or resources to cover all directions that seemed promising, we outline those here.

**Connectivity.** Training all-to-all DNA is challenging. Consequently, (and especially when scaling up) it makes sense to restrict connectivity in a way that agrees with the available hardware and makes inference more deterministic. It is also possible to slowly reduce connectivity during training by eliminating the connections between the nodes that do not interact with each other often.

**Module variety.** In this work we only studied two cases: transformer compute modules and separate attention and MLP modules. Already in this case we found that attention and MLP modules are not collapsing into transformer modules, but organize themselves as a function of depth: early layers

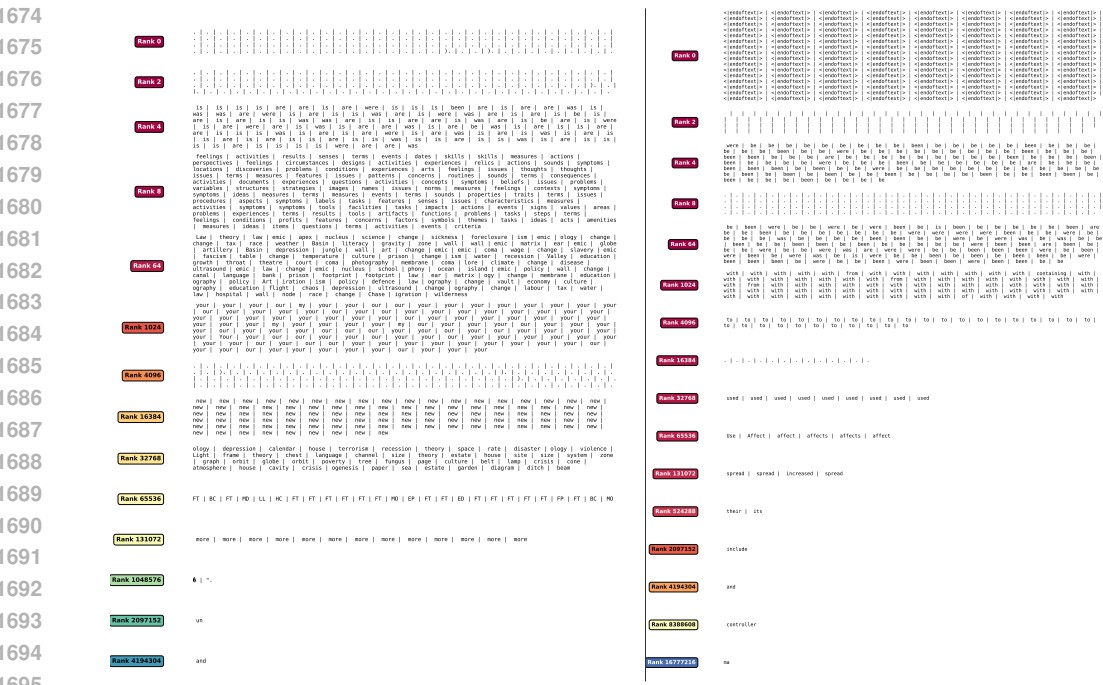

Figure 28: Tokens that go through different rank paths/ribbons. **Left**: top-1 model; **Right**: top-2 model.

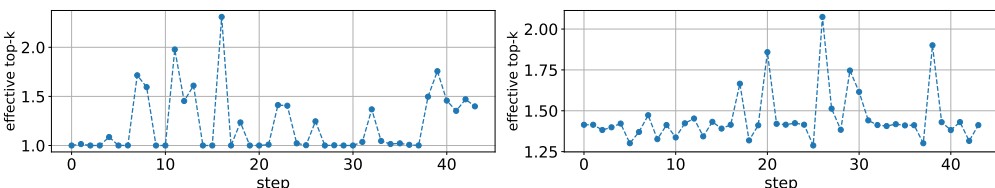

Figure 29: Effective top-$k$ for DNA-n models (language), as a function of number of forward pass steps $s$. The trend is very different from the vision models we have shown earlier. **Left**: top-1 DNA-n model. **Right**: top-2 DNA-n model.

prefer attention, while later on model prefers MLP. It makes sense to introduce a variety of modules: attention, mamba, MLP of various sizes, etc and allow the DNA to arrange them in the way it wants.

**Efficiency.** We gave a proof of principle that compute efficiency can itself be learnt from the data assuming the right incentives. It is likely that other constraints such as memory efficiency, modularity, connectivity, etc can also be learnt from the data given the right incentives.

**Complex routing.** In the present work we have focused on simple linear routers that are labeled by the step of the forward pass. We made this choice primarily to simplify training dynamics and engineering. A more natural distributed choice would assign routers directly to the nodes. This implementation is challenging especially in $k > 1$ case, but is also more in line with the distributed perspective we took here.

**Latent space reasoning.** Another advantage of DNAs is that (in principle) the model can determine itself how many steps to spend on each token. This naturally leads to a realization of latent space reasoning where the model can spend more compute on each token Geiping et al. (2025); Tack et al. (2025).

**Architectures search.** It is interesting to extract lessons from the DNAs that might be applicable to more rigid, traditional architectures. First, an interesting direction would be to introduce inhomogeneity in the model architecture. Earlier layers should be dense and (likely) not too wide, while

later layers should be much wider, and, possibly, sparse. Second, as we discuss in Appendix I: MLP and Attention do not need to be attached to each other. We train DNA models that can freely choose between MLP and attention layers. Such DNAs do not glue MLP and attention to get back the transformer architecture. Instead, they prefer to use more attention during early steps and use more MLP during later steps. We also find that the models really like weight-sharing, however it is not clear how beneficial it really is.

**MoE interpretability.** This work also suggests an interesting direction towards interpreting MoE models. Namely, one could focus on interpreting paths through the network as well as routing decisions. In fact, understanding the structure of routing matrices themselves seems like a very interesting problem.

**Data filtering.** Another interesting possibility is to flip the DNA idea backwards and design a setting where DNA plays the role of a smart data filtering system. We can imagine incentivizing the DNA model to be as memory/compute efficient as possible and then use the trained model to filter data. Given our discussion about smart ensembles, it is appropriate that a smart enough router will evolve into a natural data-filtering system because its role is to assign sub-networks of different representation power to different inputs.

