# OpenReview forum: "Towards Distributed Neural Architectures"
_ICLR.cc/2026/Conference — Submitted to ICLR 2026_

### Official Review · Reviewer_KV1h · 2025-10-31

**Soundness:** 2
**Presentation:** 3
**Contribution:** 2
**Rating:** 4
**Confidence:** 3

**Summary:**

The paper introduces Distributed Neural Architectures (DNAs), a new class of models where the computational graph is not fixed. The paper shows that DNAs, which are initialized with a set of modules (e.g., transformer, MLP) and routers, learn their own connectivity and computation patterns end-to-end. This architecture is presented as a natural generalization of sparse methods like Mixture-of-Experts (MoE) and Mixture-of-Depths (MoD). The authors train and evaluate DNAs in both the vision domain (on ImageNet, against a ViT-Small baseline) and the language domain (on Fineweb-Edu, against a GPT-2 Medium baseline).

On the positive side, the paper demonstrates that these models are competitive with their dense baselines in terms of performance. The primary contribution is a deep analysis of the emergent properties of these trained DNAs. On the negative side, there are some missing components, including a justification of the running time, and doing more in-depth analysis for vision vs. language tasks.

**Strengths:**

* The authors show that DNAs can learn to be compute-efficient by routing tokens to identity modules, and this compute allocation is interpretable (e.g., more compute is spent on visually complex images). This is interesting and seems to be novel.
* The authors also analyze the distribution of paths that tokens take (e.g., power law) and also that the learned modules can be interpretable in terms of their specializations. Frequent (low-rank) paths process high-level features like edges and flat colors, whereas infrequent (high-rank) paths are reserved for specific, low-level concepts
* The paper considers both text and visual inputs, making it more general than just studying one modality.
* The analysis extends to the language models, considering punctuation and word pieces separately.

**Weaknesses:**

* There is high computational overhead of the proposed architecture. The authors admit that the current DNA implementation "runs slower and consumes more memory" than its dense baseline counterparts. This is attributed to the "unoptimized handling of dynamically changing sequence lengths" and the inability to precompute and cache attention masks. I think this makes it difficult to appreciate the practical benefit of "compute efficiency." The savings from skipping modules are currently outweighed by the overhead of dynamic routing.
* The performance is not a clear win; the paper aims for competitiveness, but the top-1 DNA language model achieves a worse validation loss (2.754) than the standard GPT-2 baseline (2.720).
* Finally, the "fully distributed" concept is slightly compromised by empirical design choices required for stable training, such as hard-coding the first few layers as a non-routed "backbone".

**Questions:**

* It seems that optimization converges better when a hard-coded dense backbone ($N_b > 0$) is used to process all tokens before any routing begins. This seems to be a critical, un-ablated component that partially contradicts the core "fully distributed" premise. Does this finding imply that distributed routing is primarily effective as a refinement mechanism after a shared, dense feature extractor has processed the raw inputs? What happens if you try to force $N_b=0$ and compensate with other training techniques, such as a longer warmup or a different optimizer? If $N_b=0$ models fundamentally fail to converge, what does this tell us about the limitations of data-dependent routing at the earliest layers?

* Your results show a marked difference between vision and language domains. In vision, emergent parameter sharing is interpretable and correlates with image features (Fig. 7), and compute allocation is highly contextual (Fig. 5). In language, you conclude that parameter sharing is "most likely random" (Sec 4.3) and the performance gains are marginal (Table 3). Why do you believe this conceptual gap exists? Is it simply a matter of scale, where the 400M-parameter language model is too "underparametrized" for the complexity of FineWeb-Edu? Or, is it a fundamental mismatch, suggesting that a homogenous pool of Transformer blocks is less suited for language, which might require a more diverse "proto-architecture" of modules to learn meaningful, non-random specialization?

* The abstract states a token can "traverse any series of modules in any order," which implies a general, potentially cyclic, graph traversal. However, the implementation (Fig 1b, Sec 2.2) describes a "fully causal," step-based process where a router $R_s$ makes a decision at each step $s$ up to a maximum $s_{max}$. This step-based model appears to be less of a true "any order" graph and more of a deep, sequential MoE where the set of experts is simply the entire pool of modules. Could you clarify if it's possible for a token to be routed to module $M_5$ at step $s=3$ and then to module $M_2$ at step $s=4$? If so, how is this functionally different from just having two sequential routing layers? How critical is your specific $R_s$ implementation (a router-per-step) versus a more state-dependent design where each module $M_i$ has its own router that decides the next module?

---

> ### Author Response · Authors · 2025-12-03
> **Rebuttal (1/2)**
>
> We thank the reviewer for their very constructive comments and great questions!
>
> ## Weakness
>
> **Regarding Overhead**: The reviewer raises a valid point. Currently, we are not optimizing for maximum efficiency, as it is known that dynamic compute only provides benefits in extremely computation-bound scenarios, which is not presently the case. Since the purpose of this work is to better understand architecture design and create a prototype that can perform dynamic compute allocation. Scaling up the approach requires a very serious engineering and compute investment. Consequently, we left it for the time when such resources are available to us.
>
> **Performance**: As we mentioned in our draft, the purpose of this work is NOT to achieve SOTA performance, but to show that (i) self-organizing distributed architectures are trainable in the first place and (ii) to study their behavior in the largest case we could afford to train. Similar to the reason above, we leave the serious test of performances for future work.

---

> ### Author Response · Authors · 2025-12-03
> **Rebuttal (2/2)**
>
> ## Questions
>
> **Backbone**: The backbone is currently unavoidable for language modeling only. However, since it constitutes a small portion of the total parameters, we believe this does not undermine our goal of building a distributed architecture. Note here the model does not diverge but instead converges to an interpretable poor minimum, which we will elaborate on shortly. For vision tasks, since the model learns a backbone regardless, we simply include it explicitly as a general recipe.
>
> In language modeling, the backbone was introduced to prevent the model from learning a wide but shallow network topology when zero-compute modules are permitted. We believe this behavior reflects the inherent nature of language modeling: token diversity is extremely high, and it is impossible to model long sentences without understanding the meaning of each word/token. Consequently, during early training stages, the model must learn to cluster tokens as much as possible to build simple concepts, which also helps drastically reduce the loss. At the same time, depth is not needed to model complex concepts at this early stage of training. It then quickly utilizes as many modules as possible (wide) to specialize in specific simple concepts, which leaves no room for other functionalities. However, once the router has decided to skip all early steps in favor of building a wider network during this initial training phase, it becomes difficult to recover to a deeper model, ultimately leading to poor performance.
>
> Without the zero-compute expert, the model lacks this shortcut. However, our routers at the first step tend to form a dense model regardless, where we find that induction heads (always formed in early layers, see[1]) appears.  While the precise reason remains unclear, we hypothesize that the induction algorithm is sufficiently static and does not require much diversity, thus favoring a dense model structure. We believe a similar bias has been observed in many MoE models (DeepSeek v3, for example, uses dense layers at early steps, though this is rarely discussed in technical reports).
>
> We have also experimented with many different hyperparameters, including those suggested by the reviewers, but could not resolve this problem. Ultimately, we decided to use a backbone.
>
>
> [1] https://transformer-circuits.pub/2022/in-context-learning-and-induction-heads/index.html
>
>
> **Difference Between Language and Vision Tasks on Parameter Reuse/Compute Distribution**: We are not certain. Our best guess is that it's certainly a matter of scale. Our small scale language models cannot perform any difficult tasks, nor can they generate sufficiently long coherent paragraphs. In this sense, they only learn premature patterns of language and has not yet developed more interpretable internal structures.
>
> We do not believe this is an issue stemming from homogeneous pools. Comparing Figures 21 and 23 with Figure 2 (language Figures 26 and 27 with Figure 6), we find that inhomogeneous pools lead to qualitatively similar path patterns as homogeneous pools for both vision and language tasks accordingly, at least for the path distribution. Though we agree that it is entirely possible to discover interesting pattern if we carefully perform an analysis over per-module type reuse/computation patterns.
>
> **Order of Modules/Similarity to MoE**: When we say "fully causal", we mean that early tokens within the context will not attend to future tokens or been affected by them in routing decision. The scenario the reviewer described is different from what we did.
>
> From the router's perspective, this implementation is indeed similar to an MoE with a shared pool of modules. However, from the tokens' perspective, this design allows any order of passing through modules at different routing steps, which standard MoE models do not permit.
>
> The per-step router design is a choice we made for training stability. We experimented with attaching a router to each module, where decisions are made at each module rather than by a global router. However, we found that the network topology in such cases often collapses to a subset of modules and performs badly, since linear layers are not expressive enough to make different decisions across multiple steps. When using more complex routers, training stability becomes problematic, as these routers sometimes learn features themselves, subsequently causing training divergence at a later time. This is a very large space to explore and we had to leave it for the future work.

---

### Official Review · Reviewer_rCQv · 2025-11-01

**Soundness:** 3
**Presentation:** 3
**Contribution:** 3
**Rating:** 4
**Confidence:** 3

**Summary:**

The paper introduces distributed neural architectures which consists of attention, MLP, router modules. Each token or patch is learned to route among these modules end to end. The proposed architecture is competitive with dense models, and provides compute efficiency. Additionally, the routing paths of tokens among these modules is interpretable.

**Strengths:**

- Proposes a framework that generalizes most of conditional computation approaches used in training large models
- Extensive analysis of routing paths of tokens to strengthen the fact that routing is interpretable
- Results match dense baseline while being sparsely activated.
- Experiments are comprehensive across vision and language modalities.

**Weaknesses:**

- Can you provide flops taken by the proposed method? It is hard to make comparison with baseline without flop comparison as their performances are similar.  Include it in the table which presents the results for each domain.
- Learning routing is a hard problem faced in MoEs, MoDs. The proposed method doesn’t address it all, which makes the framework not useful at the current stage.
- What’s the motivation to include skip identity modules?
- Why does Top-2 DNA models always have skip modules in them and have different hidden size compared to others. It makes comparison harder.
- Can you provide details about related works as to why this approach hasn't been successful in the past?
- Can you provide specific choices made to ensure the comparison of proposed method to baseline is fair?

**Questions:**

- Intuitively, it is mentioned that the higher rank path has a high frequency of tokens, while the lower is low frequency, how exactly is the rank computed?
- The details on the architecture are somewhat unclear. Are there same modules at each step?  Is there a single router for each step?

---

> ### Author Response · Authors · 2025-12-03
>
> We thank the reviewer for their detailed comments.
>
> We want to stress that the purpose of this work is not to chase SOTA performance but rather to explore and understand what constitutes a natural network architecture and to think beyond conventional designs.
>
> ## Weaknesses
>
> **FLOPs/Model Size/Fairness**: (a) Since the context lengths are short in this case, FLOPs are equivalent to the number of (active) parameters, which we have provided in Tables 1 and 2 (and appendices). (b) The top-2 model has a different width because each token activates two modules; therefore, we chose the width differently to ensure that the active parameter counts are as close to top-1/dense models as possible without significantly changing network depth or sparsity.
>
> **Top-2 always have identities**: Table 1 is intended solely to demonstrate hyperparameter choices. In Tables 2 and 3, we include top-2 models without skip modules, which are suitable for performance comparisons.
>
> **Identity Module**: The purpose of the identity module is to allow different tokens to receive different amounts of compute, as natural data vary in difficulty and there is no reason to enforce equal computation across all tokens. Our target is to build a model that can dynamically allocate more compute to difficult tasks and less to simple ones.
>
> **Routers Are Hard to Learn**: Training instability at large scale (>8B) or nonlinear routers can be problematic, but these issues are beyond the scope of this work.
>
> **Why Our Approach Hasn't Been Successful in the Past**: To our best knowledge, no existing works have attempted this "wild" approach to training a network in the modern era. We repeatedly were told that it is not possible to train such networks. Some early works may have shared similar ideas that we were not aware of, and if they failed, we believe it was largely because the ideas were not practically implementable at the time. We also did our best to review the literature both in the main text and appendices.
>
>
> ## Questions
>
> **Path Rank Clarification**: A lower-ranked path indicates that more tokens follow that specific path during inference; similarly, a higher-ranked path indicates that fewer tokens follow it.
>
> **Details of Forward Pass/Architecture**: There are $N_m$ independent modules that can be reused at any point. Each router sees all of them and decides which module the tokens should be sent to (as we clearly state in Section 2.1, there are $N_r$ independent routers).

---

### Official Review · Reviewer_DzFA · 2025-11-02

**Soundness:** 3
**Presentation:** 4
**Contribution:** 3
**Rating:** 8
**Confidence:** 4

**Summary:**

The paper introduces a novel neural architecture design paradigm called Distributed Neural Architectures (DNAs) and exhibits them in example vision and language modeling domains. The main motivation behind this paradigm is to allow the model to, via training, discover and organize itself into sparse paths which can be executed at inference time efficiently. DNAs are not feed-forward and allow information to flow between any pair of a set of computing modules. Each computing module is chosen as a piece of a Transformer such as MLP, attention, Transformer layer itself etc. In addition, there are routing modules whose job is to route a given set of input tokens to a given set of output modules. The routing is token-choice. DNAs are trained to allocate compute dynamically. The authors train a (i) classifier in the image domain on ImageNet, (ii) a generative model in the language domain using DNA.
The general proton-architecture of DNAs is as follows:
1. Input node (embedding layer)
2. Output node (unembedding layer)
3. N_m distinct computational modules
4. N_r distinct routers

The authors bias the model towards compute efficiency by using a bias term added in the router when routing to identity modules (or skip connections basically).
They observe the model chose interpretable paths and observe the emergence of contextual compute efficiency (i.e. different tokens/patches take different compute) and input-dependent parameter sharing (reuse of modules).

For the language task, they train GPT-2 sized models (with around ~400-500M parameters) on a subset of FineWeb-edu and obtain perplexity and downstream performance numbers similar to that of Transformer baselines.

**Strengths:**

- The paper proposes a novel and interesting paradigm for training neural networks. As pointed out in the paper, the main value of the paper is in proposing the paradigm and showing that it is feasible to train performant models in this paradigm. While they do not offer SoTA performance or practical wall-clock time efficiency with current hardware, it is an important research direction.

- The proposed DNA architecture could easily have ended up much less performant than the well-optimized Transformer. But the fact that the authors manage to get it to train to be reasonably performant is surprising and an indicator of the promise of the proposed approach.

**Weaknesses:**

- Even with current Transformer architecture, we achieve a high degree of sparsity via MoEs and efficient attention layers such as Mamba or sliding window attention. In addition, we can also achieve contextual sparsity in principle with methods such as early exit (Confident Adaptive Language Modeling Schuster et. Al. 22). It is unclear if there is evidence to believe that approaches like DNA can achieve a much sparser structure than these known methods.

**Questions:**

- An important question is whether there is a fundamental reason to believe DNAs can offer more efficient pathways compared to MoEs? Can you comment on this?
- Can you also provide mode details in the paper on how you implemented training and inference for the language generative task? You mention you can’t use the standard KV caches. Are there any other optimizations that can be done instead for DNAs?

---

> ### Author Response · Authors · 2025-12-03
>
> We thank the reviewer for their encouraging comments.
>
> **Regarding the weakness and first question**: We do not have a principled argument to claim that our method could lead to more efficient pathways than MoEs/efficient attention/early exit. However, we would like to elaborate on our perspective and motivation for this work.
>
> The primary goal of this work is to demonstrate that connectivity patterns can emerge directly from data, where sparsity, efficient attention, and early exit may arise as natural outcomes of this emergence. This approach could yield fundamentally different network topologies not covered by the MoE and early exit methods the reviewer mentioned (though we acknowledge our approach is not entirely orthogonal to those methods). Note, we view this as a necessary early step towards designing a network that can be distributed across multiple datacenters in the future.
>
> Additionally, when researchers design those efficient mechanisms, the typical motivation centers on reducing computational costs, while the role of data is often overlooked. To bridge this gap, we aim to understand what preferences emerge when we allow maximum freedom in connectivity patterns, which in turn helps determine whether certain efficiency methods are inherently favored by the data and can emerge from training end-to-end. Furthermore, this flexibility offers the advantage of later distilling insights from unforeseen emergent patterns and applying efficient methods when hardware-level efficiency is required.
>
> Overall, this design provides a substantially different view for discovering better architectures in the future.
>
>
> **Regarding implementation details**: We would be happy to release the code upon receiving permission. Here, we briefly elaborate on the KV-cache challenge. The are two main difficulties:
>
> - The first challenge stems from the dynamic nature of the system: the KV-cache procedure must be adjusted since each module receives only a portion of the sequences, and this process is inherently dynamic. Consequently, each module must determine whether to update the KV-cache during each forward pass, and the KV-cache can be fragmented. This makes optimizations used in modern frameworks like vLLM more difficult to migrate to this architecture. Additionally, this approach could potentially lead to more imbalanced VRAM usage across different devices, which would require more sophisticated methods for optimization. However, we believe this can be addressed once the users have gathered the token path distribution details in most scenarios.
>
> - The other challenge is common to most recurrent models: the KV-cache grows with the amount of reuse of attention modules. While this does not increase the overall KV-cache size in our case for now, it could be a potential problem once we try to scale test time compute in the future. However, there are recent methods for reusing KV-cache in recurrent models that suggest this issue could be largely resolved in the near future.

---

### Official Review · Reviewer_tjUN · 2025-11-03

**Soundness:** 2
**Presentation:** 3
**Contribution:** 2
**Rating:** 4
**Confidence:** 4

**Summary:**

This paper proposes Distributed Neural Architectures (DNAs). DNAs use learned routers to let each token use the modules in flexible orders. Results show DNAs match dense baseline performance in vision and language tasks while learning interpretable, efficient compute allocation patterns that follow power-law distributions.

**Strengths:**

- This is an innovative work. The authors challenged the fixed architecture pipeline and proposed a genetic and flexible modular architecture.
- The authors come up with visualization designs to analyze the routing pattern of modular networks.

**Weaknesses:**

- Weak results: Evaluation results in vision tasks underperform the dense models. While the language task also show mixed results against the dense counterpart.
- Doubts on interpretability: As the authors also discovered, randomly initialized model also show some degree of "clustering". This cases doubt on the reliability of inspection the patches and read meaning from it. It may simply because the patches starts to be close to each other, and their representation remain close through out the layers, when those representation pass through the linear router, they naturally feel like having some "theme". Maybe rigorously determine whether a path have specialized on anything requires new analysis method, for example, mixing up paths from the proposed model and randomly initialized model, and then letting the researcher determine if there is any pattern without knowing the source.

**Questions:**

- Related works: https://arxiv.org/abs/2302.11529 has a lot of good related papers. Classical works like https://arxiv.org/abs/1511.02799 probably deserve a place.
- This is an innovative work. But task performance is at an disadvantage, while the interpretability is not compelling enough. Potentially the compute saving may be an advantage. To prove this point, having two dense model baselines that have the same number of param as the total number of parameters, and active parameters can be helpful.
- Regarding the vision experiments, do you have datasets in mind that particularly can benefit from adaptive computation? Maybe the method isn't inherently weaker than dense model, just the datasets are mostly homogeneous.

---

> ### Author Response · Authors · 2025-12-03
>
> We thank the reviewer for their constructive suggestions.
>
>
> **Clustering/interpretability**: The reviewer raised a very good point. Here we would like to clarify the differences between trained and random networks:
>
>
> - Random network: There is no guarantee that two similar patches $p_1$ and $p_2$ will remain close after a forward pass through a random network. This only occurs when the network is properly initialized such that the correlation between two patches can be preserved throughout [1]. More importantly, random networks capture similarities pixel by pixel. For example, a rotated patch $rot(p_1)$ might not be similar to $p_1$, though humans can tell the rotated patch conveys the same information as the original one.
>
> - Trained network: In contrast, as we demonstrated in Figure 3, trained networks cluster patches with semantic meanings. Clearly, lots of those patches are not close to each other at initialization.
>
> Furthermore, we find that the low-rank (frequently used) path at initialization does not end up as a low-rank path after training. If the reviewer's concern were valid, we would expect the low-rank path at initialization to remain a low-rank path at the end of training.
>
>
> We will further clarify this point in the updated draft.
>
>
> [1] Deep Information Propagation, https://arxiv.org/abs/1611.01232
>
>
> **Performance**: Due to limited resources, we were unfortunately unable to run a new baseline with the same number of parameters. Note that the choice we made was to make sure the number of parameters is as close as possible for different settings, while keeping the network depth.
>
> We also want to emphasize again (as mentioned in our paper) that the main focus of this work is not on achieving SOTA performance, but rather on understanding the fundamentals of neural architecture design.
>
> **About data diversity/performance**: We believe the advantage of this design can potentially show up for larger scale datasets, e.g. ImageNet 21K or larger, or potentially non-public massive datasets. Another setting worth trying would be a mixture of datasets from different disciplines. Another potential direction worth exploring would be a different training objective, for example DINO-like training objective, where the computationally intensive self-supervision training could potentially unleash more power from our more flexible architecture design.
>
> **About new references**: We thank the reviewer for pointing out these references and will add them to the related works accordingly.

---

### Meta-Review · Area_Chair_eNHR · 2025-12-19

**Summary:**

This paper looks at the problem of composing neural modules with learned routers. This is generally a difficult problem that has been tried multiple times in the community. It is particularly unfortunate this year’s ICLR situation in this particular case, as the paper would have strongly benefited from the reviewers’ discussion. I think the core of the question is: the results are mixed, but this is the largest scale such type of approach has ever been tried. Many works before have tried “similar” (in spirit, not in method) compositional architectures, and these have never performed very well, mostly due to training difficulties. This paper is clearly better (although it really should dig into the literature better; there have been many approaches on compositional architectures and neural programs, e.g., from MILA that are not cited or compared against). Ultimately, I decided to reject the paper because the main blocker for these type of approaches has always been training: it’s often unstable, slow, and requires more memory. This work suffers from the same exact limitation, unfortunately. They achieve better results than ever before by using more “modern” components, but the underlying difficulties are still there. This was a core concern of KV1h, which has not been addressed. I think the fact that “connectivity patterns can emerge directly from data” was known before, so the question about relations to MoEs are in my opinion fair.

**Reviewer Concerns:**

I think the rebuttal didn't really move the needle. rCQv was not a particularly informative review, so I have largely discarded it.

tjUN had concerns on the performance and related works, which I largely share and find they have not been addressed by the authors. I also did not appreciate their response on the related works. The reviewer essentially said that the paper had positioning issues wrt related works and that the paper didn't do justice to the long body of works that happened before. This is kind of important as the paper claims this is the first time these approaches work, and that neural connectivity has never been learned from data before. A better response would have been from the authors to go do a deepdive and come back with a serious literature review. The references mentioned by the reveiwers were not meant to be exhaustive, but a pointer to show the authros did not take the literaterature review seriously enough.

The other concerns were about performance and MoEs, which have not been addressed well.

**Reviewer Scores:**

I don't think the reviewers would have significantly changed their scores.

---

### Decision · Program_Chairs · 2026-01-26

Reject